# FUSED-PLANES: WHY TRAIN A THOUSAND TRI-PLANES WHEN YOU CAN SHARE?

**Karim Kassab**[*1,2]**, Antoine Schnepf**[*1,3]**, Jean-Yves Franceschi**[1]**, Laurent Caraffa**[2]**, Flavian Vasile**[1]**, Jeremie Mary**[1]**, Andrew Comport**[†3]**, Valérie Gouet-Brunet**[†2]

[*†] Equal contribution

[1] Criteo AI Lab, Paris, France
[2] LASTIG, Université Gustave Eiffel, IGN-ENSG, F-94160 Saint-Mandé
[3] Université Côte d'Azur, CNRS, I3S, France

## ABSTRACT

Tri-Planar NeRFs enable the application of powerful 2D vision models for 3D tasks, by representing 3D objects using 2D planar structures. This has made them the prevailing choice to model large collections of 3D objects. However, training Tri-Planes to model such large collections is computationally intensive and remains largely inefficient. This is because the current approaches independently train one Tri-Plane per object, hence overlooking structural similarities in large classes of objects. In response to this issue, we introduce Fused-Planes, a novel object representation that improves the resource efficiency of Tri-Planes when reconstructing object classes, all while retaining the same planar structure. Our approach explicitly captures structural similarities across objects through a latent space and a set of globally shared base planes. Each individual Fused-Planes is then represented as a decomposition over these base planes, augmented with object-specific features. Fused-Planes showcase state-of-the-art efficiency among planar representations, demonstrating $7.2\times$ faster training and $3.2\times$ lower memory footprint than Tri-Planes while maintaining rendering quality. An ultra-lightweight variant further cuts per-object memory usage by $1875\times$ with minimal quality loss. Our project page can be found at https://fused-planes.github.io .

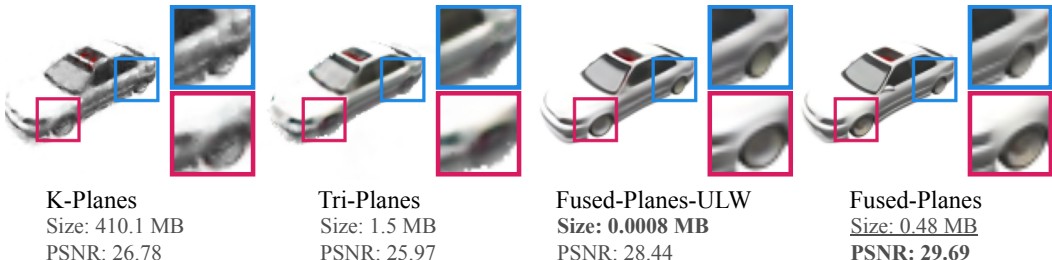

| K-Planes | Tri-Planes | Fused-Planes-ULW | Fused-Planes |
|---|---|---|---|
| Size: 410.1 MB | Size: 1.5 MB | **Size: 0.0008 MB** | Size: 0.48 MB |
| PSNR: 26.78 | PSNR: 25.97 | PSNR: 28.44 | **PSNR: 29.69** |

Figure 1: **Comparison of planar representations under the same budget.** Our method achieves the best rendering quality and the best memory footprint among planar representations when training large classes of 3D objects under a fixed time budget (7 minutes per object in this illustration). Fused-Planes-ULW designates the ultra-lightweight variant of Fused-Planes.

## 1 INTRODUCTION

Tri-planar representations (Chan et al., 2022; Fridovich-Keil et al., 2023) have recently driven significant progress in 3D computer vision, offering a unique advantage: they model 3D objects while remaining interpretable as 2D structures due to their planar format. This planarity makes them compatible with standard image-based models (e.g. CNNs), thereby unlocking new ways 2D vision models can be used for 3D tasks (Hong et al., 2024; Anciukevičius et al., 2023; Mercier et al.,

2025). Given that such applications are inherently data-intensive, the need to train large collections of Tri-Planes for 3D reconstruction has become increasingly prevalent (Cardace et al., 2024; Shue et al., 2023; Ju & Li, 2025), and a costly preliminary step in 3D research (Liu et al., 2024; Wang et al., 2023, Sections 4.1 and 5). Yet, most existing methods overlook this costly 3D reconstruction step, focusing instead on the downstream tasks that planar representations enable. As such, using planar representations for large-scale 3D reconstruction remains largely suboptimal in terms of resource efficiency, since existing methods train each Tri-Plane independently, ignoring the structural similarities that often exist across large object classes. This oversight leads to redundant computations and inefficient memory usage. As a result, constructing a dataset of Tri-Planes is currently unnecessarily computationally intensive.

In this work, we address the challenges associated with the computationally expensive task of large-scale 3D reconstruction using planar methods. We introduce Fused-Planes, a novel tri-planar representation that efficiently models large classes of 3D objects. Fused-Planes effectively reduces the resource costs associated with Tri-Planes by leveraging the structural similarities shared across multiple objects. Additionally, Fused-Planes retains the planar property of Tri-Planes that has enabled their integration into existing pipelines, and thus retains their compatibility with recent approaches.

**First,** our Fused-Planes split an object representation into two separate components: the first "Micro" component learns features specific to the object at hand; the second "Macro" component is a learned decomposition over a set of base planes, where each base plane encapsulates structural similarities across the class of objects we want to reconstruct. **Second**, we train Fused-Planes with a 3D-aware latent space (Schnepf et al., 2025), which provides a continuous and structured representation of objects, and accelerates the rendering and training of Fused-Planes.

The combination of these two cost-reducing components is essential. On the one hand, the latent space provides a more effective representation for disentangling object-specific details from class-level structural similarities, making it easier to capture these similarities with the set of base planes. On the other hand, the micro-macro decomposition is essential to eliminate the quality losses associated with using a latent space.

We conduct extensive experiments justifying these design choices and comparing our method with current planar representations when training on large classes of objects. Fused-Planes presents $7.2\times$ faster training than Tri-Planes, while requiring $3.2\times$ less memory footprint and retaining a similar rendering quality, thus establishing a new state-of-the-art in efficiency for planar scene representations. Moreover, an ultra-lightweight variant of Fused-Planes trades off minor rendering quality for substantial gains in memory footprint: $1875\times$ less than Tri-Planes. To the best of our knowledge, our work is the first to improve upon the resource efficiency of Tri-Planes.

## 2 RELATED WORK

**Tri-Planes.** Tri-Planes (Chan et al., 2022) are widely used for modeling large collections of 3D objects and have attracted considerable attention due to their seamless integration with standard image-based models. In recent works, Tri-Planes are commonly used within a framework that involves solving two main tasks (Shue et al., 2023; Ju & Li, 2025). The first task is large-scale **3D reconstruction**, which consists of training Tri-Planes to properly model a large set of 3D objects. Once this prerequisite task is completed, the Tri-Planes can be reshaped into 2D image-like tensors, an operation made possible by their planar structure, making them easily integrable with image-based models. Once trained and reshaped, Tri-Planes are applied to a second, **targeted task**, in conjunction with a chosen image-based model. While recent studies have focused heavily on exploring diverse targeted tasks such as editing (Ki et al., 2025), classification (Cardace et al., 2024), generation (Liu et al., 2024), and feed-forward reconstruction (Wang et al., 2023), the first large-scale reconstruction task itself remains inefficient and sub-optimal, which has inspired our research direction. A more detailed discussion of works using Tri-Planes for downstream tasks can be found in Appendix (Section A).

**Compatibility of NeRF methods with image-based models.** Since NeRF (Mildenhall et al., 2020), methods such as Instant-NGP (Müller et al., 2022), TensoRF (Chen et al., 2022), and 3D Gaussian Splatting (Kerbl et al., 2023, 3DGS) have greatly advanced single-scene reconstruction. However, unlike Tri-Planes, these representations cannot be directly reshaped into image-like ten-

sors. Recent works attempt to work around this by converting 3DGS scenes into 2D maps using various parametrization techniques (e.g. encoding a gaussian per pixel (Li et al., 2024)). However, this explicit parameterization either (i) requires 3D-to-2D unwrapping techniques (e.g., UV maps (Hu et al., 2024; Pang et al., 2024) or Morton-order mappings (Jiang et al., 2025)) to preserve spatial semantics across different sides of an object, or (ii) damage image-like spatial semantics, since adjacent pixels may correspond to spatially distant Gaussians (Szymanowicz et al., 2024). In contrast, Tri-Planes require no such preprocessing. They are fixed-size, fully structured, and consistent across scenes, making them directly compatible with standard image-based architectures.

These properties have made them the most adopted approach in large-scale 3D reconstruction, and motivate our focus on adopting *purely* planar representations, without introducing auxiliary non-planar components (Wu et al., 2024a).

Our work aims to address the inefficiencies of large-scale 3D reconstruction with Tri-Planes. **First**, we design Fused-Planes to be a tri-planar *shared representation* that captures the structural similarities in object classes. **Second**, we train Fused-Planes as *latent NeRFs*, facilitating the learning of our shared representations. These design choices lead to substantial reductions in both training time and memory footprint.

**Shared representations.** Shared representations denote approaches that model multiple objects by utilizing common components. These representations encode an abstraction of a set of objects, effectively capturing dataset-level information such as structural similarities and differences among objects. For example, Jang & Agapito (2021) represent multiple objects of the same class within a single NeRF (MLP) by conditioning it on distinct latent codes for shape and appearance, which allows shape and appearance to be edited independently. Similarly, Schwarz et al. (2021); Niemeyer & Geiger (2021) adopt a shared representation implemented within a GAN framework, which enables the generation of novel objects and scenes. Notably, shared representations have been employed to reduce memory footprint when modeling multiple 3D objects. For instance, Singh et al. (2024) encode multiple scenes into a single NeRF using learned pseudo-labels, thereby reducing memory footprint. However, their method cannot scale beyond 20 scenes. Our work also utilizes shared representations for resource efficiency, but remains scalable to thousands of objects while reducing both memory footprint and training time. To the best of our knowledge, our method is the first to explicitly integrate shared representations with planar structures.

**Latent NeRFs.** Latent NeRFs involve training neural scene representations within the latent space of an image autoencoder, rather than directly using raw RGB images. Several recent works have utilized Latent NeRFs for 3D generation (Metzer et al., 2023; Seo et al., 2023; Ye et al., 2023; Chan et al., 2023), scene editing (Khalid et al., 2023; Park et al., 2024), and scene reconstruction (Aumentado-Armstrong et al., 2023) with improved quality. Recently, Schnepf et al. (2025) employed latent NeRFs to accelerate NeRF training. Their approach enables training various NeRF architectures within a 3D-aware latent space, resulting in substantial speed-ups but at the expense of a notable degradation in rendering quality. Our work builds upon Schnepf et al. (2025) by training our proposed Fused-Planes representation in a 3D-aware latent space. However, unlike Schnepf et al. (2025) who pre-train a generic latent space for all NeRF representations, we train the 3D-aware latent space jointly with our scene representations, which proves essential for preserving rendering quality. This improvement enables us to achieve substantial speed-ups without quality compromises.

## 3 METHOD

Our method efficiently reconstructs large collections of 3D objects using tri-planar representations. Section 3.1 presents our novel Fused-Planes representation, which splits an object representation into an object-specific "micro" component and a "macro" component derived from shared base representations. These base representations are trained on the entire dataset, allowing to capture global structural patterns shared by the objects being reconstructed. To achieve this, we train the set of Fused-Planes in a jointly learned 3D-aware latent space, which encodes the target objects in a compact and well-structured space, thereby facilitating the learning of shared patterns with our base planes. Section 3.2 describes our training procedure for Fused-Planes and the 3D-aware latent space. Figure 2 presents an overview of our method.

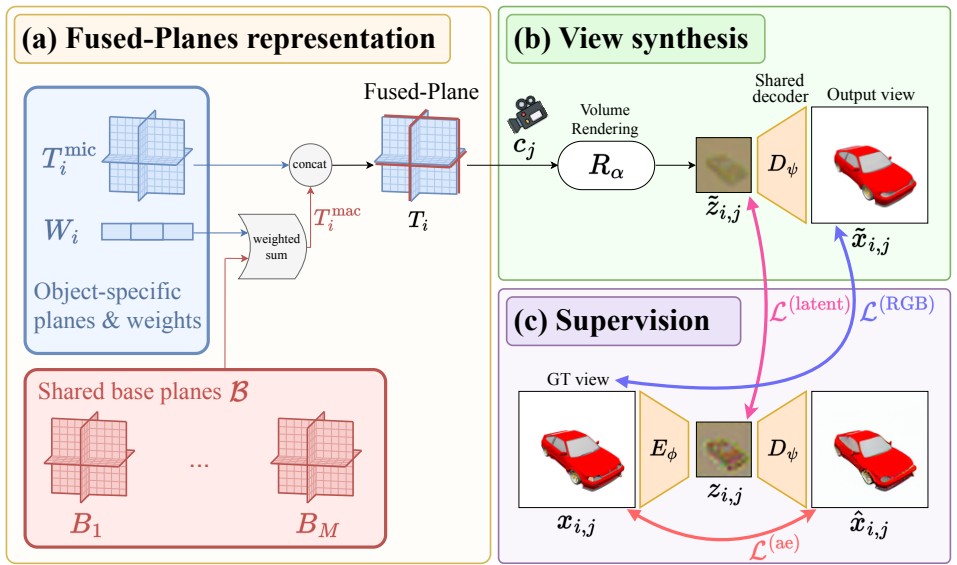

Figure 2: **Method overview.** A set of Fused-Planes $\{T_i\}$ reconstructs a class of 3D objects $\{O_i\}$ from their GT views $\{x_{i,j}\}$, where $i$ and $j$ respectively denote the object and the view indices. For clarity, only one Fused-Planes is shown. **(a)** Each Fused-Planes $T_i$ is formed from a micro plane $T_i^{\text{mic}}$ which captures object-specific information, and a macro plane $T_i^{\text{mac}}$ computed via a weighted summation over a set of shared base planes $\mathcal{B}$. This base captures class-level information like structural similarities across objects. **(b)** View synthesis is performed in the latent space of an auto-encoder ($E_\phi$, $D_\psi$) via classical volume rendering. The rendered latent image $\tilde{z}_{i,j}$ (low resolution) is decoded to obtain the output RGB view (high resolution). **(c)** The Fused-Planes components (i.e. $T_i^{\text{mic}}$, $\mathcal{B}$, $W_i$) and the autoencoder are supervised with three reconstructive losses.

**Notation.** We denote $\mathcal{O} = \{O_1, ..., O_N\}$ a large set of $N$ objects drawn from a common distribution. Each object $O_i = \{(x_{i,j}, c_{i,j})\}_{j=1}^{V}$ consists of $V$ posed views. Here, $x_{i,j}$ and $c_{i,j}$ respectively denote the $j$-th view and camera pose of the $i$-th object $O_i$. We denote $\mathcal{T} = \{T_1, ..., T_N\}$ the set of Fused-Planes representations modeling the objects in $\mathcal{O}$.

### 3.1 FUSED-PLANES ARCHITECTURE.

**Pre-requisite: Tri-Planes.** Tri-Plane representations (Chan et al., 2022) are explicit-implicit scene representations enabling scene modeling in three axis-aligned orthogonal feature planes, each of resolution $K \times K$ with feature dimension $F$. To query a 3D point $x \in \mathbb{R}^3$, it is projected onto each of the three planes to retrieve bilinearly interpolated feature vectors. These feature vectors are then aggregated via summation and passed into a small neural network with parameters $\alpha$ to retrieve the color and density, which are then used for volume rendering (Kajiya & Von Herzen, 1984).

Notably, Tri-Planes can be represented as 2D structures by reshaping them into $K \times K$ images with $3F$ channels. As such, they can be seamlessly integrated in image-based pipelines. This planar property has been fundamental for their widespread adoption, and it is preserved in Fused-Planes.

**Architecture of a Fused-Planes.** Fused-Planes is a novel planar 3D representation that builds upon Tri-Planes. A Fused-Planes splits a planar representation into object-specific features, and class-level features, which allows to learn common structures across the large set of objects. Specifically, a Fused-Planes representation $T_i$ of object $O_i$ is composed of a "micro" plane $T_i^{\text{mic}}$ integrating object-level information, and a "macro" plane $T_i^{\text{mac}}$ that encompasses class-level information:

$$T_i = T_i^{\text{mic}} \oplus T_i^{\text{mac}} , \qquad (1)$$

where $\oplus$ concatenates two planar structures along the feature dimension. We denote by $F^{\text{mic}}$ the dimensionality of local features in $T_i^{\text{mic}}$ and by $F^{\text{mac}}$ the dimensionality of global features in $T_i^{\text{mac}}$, with the total dimensionality of features in $T_i$ being $F = F^{\text{mic}} + F^{\text{mac}}$.

The micro planes $T_i^{\mathrm{mic}}$ are object-specific, and are hence independently learned for every object. The macro planes $T_i^{\mathrm{mac}}$ represent globally captured information that is relevant for the current object. They are computed for each object from shared base planes $\mathcal{B} = \{B_k\}_{k=1}^M$ by the weighted sum:

$$T_i^{\mathrm{mac}} = W_i\mathcal{B} = \sum_{k=1}^{M} w_i^k B_k \ , \tag{2}$$

where $W_i$ are learned coefficients for object $O_i$. The base of planes $\{B_k\}_{k=1}^M$ is shared among objects and capture class-level structural similarities. With this approach, the number of micro planes is equal to the number of objects $N$, while the number of macro planes $M$ is a constant hyper-parameter. We take $M > 1$ in order to capture diverse information, which our experiments showed to be beneficial for maintaining rendering quality, and $M \ll N$. Overall, decomposing Fused-Planes into micro and macro components reduces the number of trainable features per-object compared to traditional Tri-Planes, thus accelerating training and reducing total memory footprint.

**Fused-Planes-ULW.** We propose an ultra-lightweight (ULW) variant of our method with $F^{\mathrm{mic}} = 0$ (only macro planes), where we achieve substantial savings in memory footprint at the expense of a slight reduction in rendering quality.

**3D-aware latent space.** While Tri-Planes are traditionally used to model objects in the RGB space, we train Fused-Planes in the latent space of an image autoencoder, defined by an encoder $E_\phi$ and a decoder $D_\psi$. This is because a high-dimensional RGB space lacks structure, making it poorly suited for effectively capturing structural similarities. In contrast, a 3D-aware latent space (Schnepf et al., 2025) provides a structured and continuous encoding of the objects, which is, as proven by our ablations, more suited for disentangling structural similarities from object-specific details. Additionally, this latent space allows for a reduced rendering resolution, which alleviates the cost of volume rendering and contributes to accelerating our training. In practice, we train our 3D-aware latent space jointly with our Fused-Planes, which tailors it specifically for our decomposed object representation.

At inference, given a camera pose $c_j$, we render a latent Fused-Plane $T_i$ as follows:

$$\tilde{z}_{i,j} = R_\alpha(T_i, c_j) \ , \qquad\qquad \tilde{x}_{i,j} = D_\psi(\tilde{z}_{i,j}) \ , \tag{3}$$

where $R_\alpha$ is the Fused-Plane renderer with trainable parameters $\alpha$, $\tilde{z}_{i,j}$ is the rendered latent image, and $\tilde{x}_{i,j}$ is the corresponding RGB decoded rendering.

## 3.2 Training a Large Set of Fused-Planes

This section outlines our training strategy to learn a large set of objects. In brief, we jointly train the set of Fused-Planes and the 3D-aware latent space. Figure 2 provides an overview of our pipeline.

**Training a set of Fused-Planes jointly with the 3D-aware latent space.** We train the set of Fused-Planes $\mathcal{T}$ to reconstruct the set of objects $\mathcal{O}$ from posed views. As described above, we conduct this training in a 3D-aware latent space in a joint manner. To do so, we adapt the 3D regularization losses from Schnepf et al. (2025). Note that our 3D-aware latent space differs from the one in (Schnepf et al., 2025), as it is subject to an additional training constraint coming from our micro-macro decomposition. This allows us to obtain a latent space that is not only 3D-aware, but also adapted to our Fused-Planes representations.

We supervise a Fused-Planes $T_i$ and the encoder $E_\phi$ in the latent space with the loss $\mathcal{L}^{(\mathrm{latent})}$:

$$\mathcal{L}_{i,j}^{(\mathrm{latent})}(\phi, T_i) = \|z_{i,j} - \tilde{z}_{i,j}\|_2^2 \ , \tag{4}$$

where $z_{i,j} = E_\phi(x_{i,j})$ is the encoded ground truth image, $\tilde{z}_{i,j} = R_\alpha(T_i, c_{i,j})$ is the rendered latent image, and $T_i = T_i^{\mathrm{mic}} \oplus T_i^{\mathrm{mac}}$. This loss optimizes the encoder parameters and the Fused-Planes parameters to align the encoded latent images $z_{i,j}$ and the rendering $\tilde{z}_{i,j}$. We also supervise $T_i$ and the decoder $D_\psi$ in the RGB space via $\mathcal{L}^{(\mathrm{RGB})}$:

$$\mathcal{L}_{i,j}^{(\mathrm{RGB})}(\psi, T_i) = \|x_{i,j} - \tilde{x}_{i,j}\|_2^2 \ , \tag{5}$$

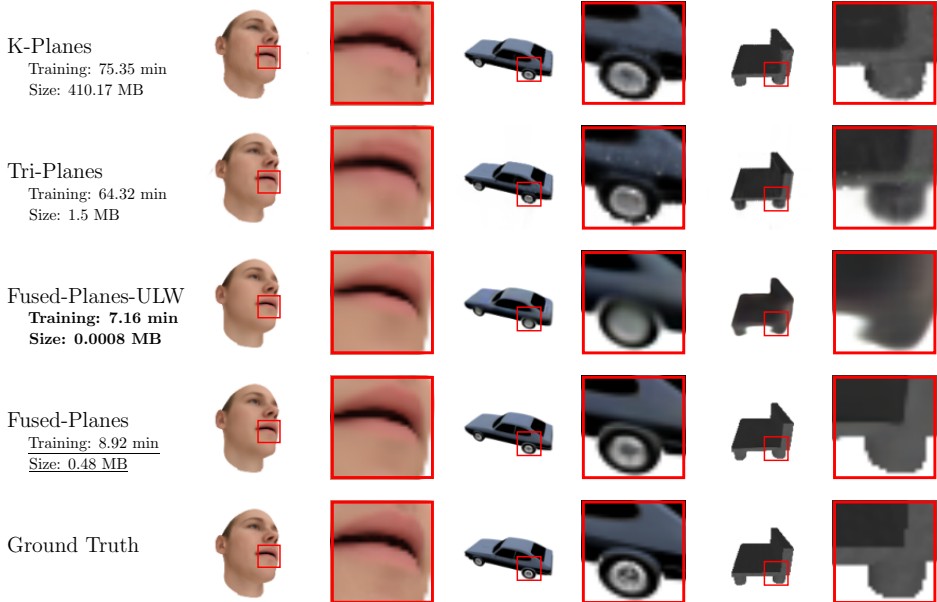

Figure 3: **Qualitative comparison.** We show comparisons of our method with other planar scene representations for NVS on held-out test views. Our method achieves the fastest training with the lowest memory footprint, while maintaining a comparable rendering quality.

where $x_{i,j}$ is the ground truth image, and $\tilde{x}_{i,j} = D_\psi(\tilde{z}_{i,j})$ is the decoded rendering. This loss ensures a good rendering quality when decoded to the RGB space, and finds the optimal decoder for this task. Finally, we adopt the reconstructive objective $\mathcal{L}^{(\text{ae})}$ supervising the auto-encoder:

$$\mathcal{L}_{i,j}^{(\text{ae})}(\phi, \psi) = \|x_{i,j} - \hat{x}_{i,j}\|_2^2 \, , \tag{6}$$

where $\hat{x}_{i,j} = D_\psi(E_\psi(x_{i,j}))$ is the reconstructed ground truth image.

Overall, our full training objective is composed of the three previous losses summed over $\mathcal{O}$ to optimize the set of Fused-Planes $\mathcal{T}$, the encoder $E_\phi$, and the decoder $D_\psi$:

$$\min_{\mathcal{T}, \phi, \psi} \sum_{i=1}^{N} \sum_{j=1}^{V} \lambda^{(\text{latent})} \mathcal{L}_{i,j}^{(\text{latent})}(\phi, T_i) + \lambda^{(\text{RGB})} \mathcal{L}_{i,j}^{(\text{RGB})}(\psi, T_i) + \lambda^{(\text{ae})} \mathcal{L}_{i,j}^{(\text{ae})}(\phi, \psi) \, , \tag{7}$$

where $\lambda^{(\text{latent})}$, $\lambda^{(\text{RGB})}$, and $\lambda^{(\text{ae})}$ are hyper-parameters.

By the end of this training, the set of Fused-Planes $\mathcal{T}$ including the base planes $\mathcal{B}$ are learned and effectively model the objects in $\mathcal{O}$. Additional object representations could still be trained by utilizing the frozen shared components. For more implementation details, we refer the reader to the Appendix (Section B and Algorithm 1).

## 4 EXPERIMENTS

**Task.** As discussed in Section 2, our goal is to reduce the resource costs of planar representations in large-scale 3D modeling. To establish the practical utility of our representation, it must satisfy two criteria: (i) accurately represent the types of 3D objects typically modeled with Tri-Planes, and (ii) demonstrate competitive resource efficiency relative to the Tri-Planes baseline. Regarding 3D modeling performance, we adopt the standard evaluation protocol for 3D representations and assess our method on the task of 3D reconstruction via Novel View Synthesis (NVS). For resource efficiency, we measure the per-object training time and memory footprint when modeling large object classes.

Table 1: **Comparison with planar methods.** Fused-Planes reduces the quality gap between Tri-Planes and K-Planes, while requiring three orders of magnitude less memory footprint, and having a significantly faster training, thus establishing a new state-of-the-art in efficiency for modeling large object classes with planar representations.

| | Planar | Training (min) | Size (MB) | ShapeNet datasets | | | Basel Faces | | |
|---|---|---|---|---|---|---|---|---|---|
| | | | | PSNR | SSIM | LPIPS | PSNR | SSIM | LPIPS |
| K-Planes (Fridovich-Keil et al., 2023) | ✓ | 75.35 | 410.17 | **30.88** | 0.956 | 0.043 | **40.23** | **0.991** | **0.005** |
| Tri-Planes (Chan et al., 2022) | ✓ | 64.32 | 1.50 | 28.15 | 0.919 | 0.121 | 36.47 | 0.980 | 0.013 |
| Fused-Planes-ULW (ours) | ✓ | **7.16** | **0.0008** | 29.02 | 0.937 | 0.092 | 33.96 | 0.955 | 0.010 |
| Fused-Planes (ours) | ✓ | 8.96 | 0.48 | 30.47 | **0.957** | **0.042** | 37.24 | 0.973 | 0.006 |

Table 2: **Comparison with methods using shared representations.** Fused-Planes demonstrates more favorable NVS quality and per-object resources requirements compared to CodeNeRF.

| | Planar | Training (min) | Size (MB) | ShapeNet datasets (avg) | | | Basel Faces | | |
|---|---|---|---|---|---|---|---|---|---|
| | | | | PSNR | SSIM | LPIPS | PSNR | SSIM | LPIPS |
| CodeNeRF (Jang & Agapito, 2021) | ✗ | 14.96 | 0.0026 | 28.34 | 0.930 | 0.121 | 35.46 | 0.972 | 0.010 |
| CodeNeRF-A (our adaptation) | ✗ | 9.54 | 0.0026 | 26.99 | 0.915 | 0.126 | 35.44 | 0.971 | 0.010 |
| Fused-Planes-ULW (ours) | ✓ | **7.16** | **0.0008** | 29.02 | 0.937 | 0.092 | 33.96 | 0.955 | 0.010 |
| Fused-Planes (ours) | ✓ | 8.96 | 0.48 | **30.47** | **0.957** | **0.042** | **37.24** | **0.973** | **0.006** |

Table 3: **Report of standard NeRF methods.** To provide the reader with a broader perspective, we report the metrics of other well-established NeRF methods. As discussed, these methods do not share the same architectural versatility as planar methods and are designed for different objectives.

| | Planar | Training (min) | Size (MB) | ShapeNet datasets (avg) | | | Basel Faces | | |
|---|---|---|---|---|---|---|---|---|---|
| | | | | PSNR | SSIM | LPIPS | PSNR | SSIM | LPIPS |
| Vanilla-NeRF (Mildenhall et al., 2020) | ✗ | 636.8 | 22.00 | 35.85 | 0.977 | 0.027 | 42.91 | **0.996** | **0.001** |
| Instant-NGP (Müller et al., 2022) | ✗ | 7.52 | 189.13 | 34.06 | 0.973 | 0.022 | 36.54 | 0.981 | 0.009 |
| TensoRF (Chen et al., 2022) | ✗ | 68.93 | 208.32 | **36.74** | **0.985** | **0.013** | 40.66 | 0.991 | 0.004 |
| 3DGS (Kerbl et al., 2023) | ✗ | 9.37 | 27.66 | 32.95 | 0.975 | 0.043 | **43.06** | 0.995 | 0.002 |
| Fused-Planes-ULW | ✓ | **7.16** | **0.0008** | 29.02 | 0.937 | 0.092 | 33.96 | 0.955 | 0.010 |
| Fused-Planes | ✓ | 8.96 | 0.48 | 30.47 | 0.957 | 0.042 | 37.24 | 0.973 | 0.006 |

**Evaluation Protocol** To evaluate the NVS quality of the learned objects, we compute the PSNR ($\uparrow$), SSIM ($\uparrow$) and LPIPS (Zhang et al., 2018, $\downarrow$) between never-seen reference views and corresponding NVS views. To evaluate the resource requirements, we report per-object training time, per-object memory footprint (excluding shared components), and total memory footprint. Training times are measured using a single NVIDIA L4 GPU.

**Baselines.** We compare Fused-Planes with three distinct lines of work. **First,** the central comparison is with planar scene representations, specifically Tri-Planes (Chan et al., 2022) and K-Planes (Fridovich-Keil et al., 2023), the only current works having planar structures. Tri-Planes is our baseline architecture and hence is our main point of comparison. K-Planes extend Tri-Planes to improve rendering quality by utilizing multi-scale planes, sacrificing on memory footprint, and most importantly the explicit 2D structure, as multi-scale planes cannot be directly reshaped into a single 2D structure. **Second,** we compare Fused-Planes with works utilizing shared representations. From this category, we consider CodeNeRF (Jang & Agapito, 2021), a recent non-planar method utilizing a shared NeRF conditioned by latent vectors. We also compare with CodeNeRF-A, our adaptation of CodeNeRF designed to improve efficiency (more details in Section F). Note that we do not compare with C3-NeRF (Singh et al., 2024) as their approach does not scale beyond 20 scenes. **Third,** and to provide the reader with a larger perspective, we report the performance of other well-established non-planar scene representations (Mildenhall et al., 2020; Müller et al., 2022; Chen et al., 2022; Kerbl et al., 2023), using their Nerfstudio (Tancik et al., 2023) implementations. While these methods are designed to model scenes *individually* with high fidelity, they are not as readily integrable with image-based models as planar methods, and therefore lack their architectural versatility. As such, they are not our primary point of comparison, but are included to provide broader context.

Table 4: **Multi-class training.** The first three rows correspond to single-class training, where a separate Fused-Planes model is trained for each individual class. The remaining rows report the results of multi-class training, where a single Fused-Planes model is trained jointly on multiple classes. The results show that Fused-Planes is applicable to multi-class data and continues to outperform Tri-Planes in this setting.

| | # Classes | Speakers | | | Sofas | | | Furniture | | | Cars | | |
|---|---|---|---|---|---|---|---|---|---|---|---|---|---|
| | | PSNR | SSIM | LPIPS | PSNR | SSIM | LPIPS | PSNR | SSIM | LPIPS | PSNR | SSIM | LPIPS |
| Fused-Planes | 1 | 29.99 | 0.953 | 0.053 | 30.92 | 0.958 | 0.028 | 30.72 | 0.960 | 0.053 | 30.27 | 0.960 | 0.033 |
| Fused-Planes-ULW | 1 | 29.22 | 0.941 | 0.087 | 29.02 | 0.931 | 0.084 | 29.14 | 0.933 | 0.142 | 28.71 | 0.943 | 0.055 |
| Tri-Planes | 1 | 27.02 | 0.909 | 0.134 | 28.48 | 0.921 | 0.103 | 27.42 | 0.894 | 0.210 | 29.69 | 0.953 | 0.036 |
| Fused-Planes | 2 | 30.03 | 0.953 | 0.053 | 30.37 | 0.953 | 0.033 | — | — | — | — | — | — |
| Fused-Planes-ULW | 2 | 28.63 | 0.925 | 0.108 | 28.93 | 0.931 | 0.085 | — | — | — | — | — | — |
| Fused-Planes | 3 | 29.84 | 0.952 | 0.053 | 30.08 | 0.950 | 0.034 | 30.31 | 0.955 | 0.064 | — | — | — |
| Fused-Planes-ULW | 3 | 29.30 | 0.937 | 0.088 | 29.33 | 0.937 | 0.064 | 29.47 | 0.942 | 0.118 | — | — | — |
| Fused-Planes | 4 | 29.72 | 0.951 | 0.055 | 29.70 | 0.948 | 0.038 | 29.79 | 0.951 | 0.073 | 29.15 | 0.952 | 0.040 |
| Fused-Planes-ULW | 4 | 28.12 | 0.924 | 0.110 | 28.34 | 0.923 | 0.084 | 28.54 | 0.927 | 0.154 | 27.73 | 0.933 | 0.074 |

**Datasets & Experimental Details.** We evaluate our method on large-scale 3D data. Consistently with Tri-Planes and CodeNeRF, we use ShapeNet (Chang et al., 2015), from which we take four categories: Cars, Furniture, Speakers and Sofas. Additionally, we adopt the large-scale front-facing Basel-Face dataset (Paysan et al., 2009). More dataset details can be found in Section C. In our experiments, we train a set of Fused-Planes to reconstruct $N = 2000$ objects. We use planes of dimensionality $K \times K \times F$, where $K = 64$ and $F = 32$ for all planar representations. For Fused-Planes, we take $F^{\mathrm{mic}} = 10$, $F^{\mathrm{mac}} = 22$, and $M = 50$. For Fused-Planes-ULW, we take $F^{\mathrm{mic}} = 0$, $F^{\mathrm{mac}} = 32$, and $M = 50$. We detail our hyper-parameters in Section G. We adopt the pre-trained VAE from Stable Diffusion (Rombach et al., 2022) as initialization for our VAE.

## 4.1 MAIN RESULTS

Main results appear in Tables 1 to 3 and Figures 1 and 3. Detailed results are available in Section D.

Compared to other **planar scene representations** (Figure 1 and Table 1), Fused-Planes exhibits a significant reduction in resource costs, demonstrating $7.2\times$ faster training and $3.2\times$ less memory footprint than Tri-Planes, and $8.4\times$ faster training and $854\times$ less memory footprint than K-Planes. It improves rendering quality over Tri-Planes while reducing the gap with K-Planes, but without K-Planes' orders-of-magnitude higher memory cost or multi-scale complexity. Fused-Planes-ULW trades off minor rendering quality for substantial gains in memory footprint: one object requires $1875\times$ less memory footprint than Tri-Planes, and $512\,000\times$ less memory footprint than K-Planes. Furthermore, Figure 4 illustrates the evolution of the resource requirements as the number of objects increases. Moreover, a detailed breakdown of the memory footprint of Fused-Planes can be found in the Appendix (Table 15). All in all, Fused-Planes establishes a new state-of-the-art in terms of resource efficiency for planar scene representations.

As for other methods utilizing **shared representations** (Table 2), Fused-Planes and Fused-Planes-ULW showcase up to $2\times$ faster training times, and an improved rendering quality. Fused-Planes-ULW also requires less memory footprint per-object.

For broader context, we report results on other well-established **non-planar** NeRF methods (Table 3 and Figure 5). Fused-Planes, like all other planar representations, showcases lower rendering quality, which is an acceptable trade-off as planar methods have a different primary objective (Section 2).

## 4.2 RESULTS ON MULTI-CLASS TRAINING

Table 4 reports a set of experiments in which Fused-Planes and Fused-Planes-ULW are trained on datasets containing scenes from multiple object classes. Specifically, we introduce three new datasets that combine two, three, and four object classes, composed from our initial object categories. Specific details about the construction of these datasets are available in the Appendix (Section C). The results demonstrate that Fused-Planes is applicable to multi-class data and continues to outperform Tri-Planes in this setting. Furthermore, a minor reduction in quality appears as more classes are included, reflecting the increased scene diversity that shared base planes must capture.

Table 5: **Ablation study.** Comparison of NVS quality and per-object resource costs for different ablations of our method on ShapeNet Cars. Fused-Planes outperforms all of its ablations. Fused-Planes-ULW trades off minor NVS quality for substantial savings in memory footprint.

| | Latent Space | Micro Planes | Macro Planes | Training (min) | Size (MB) | PSNR | SSIM | LPIPS |
|---|---|---|---|---|---|---|---|---|
| Fused-Planes ($M = 1$) | ✓ | ✓ | ✓ | 8.48 | 0.48 | 27.69 | 0.942 | 0.042 |
| Fused-Planes (Micro) | ✓ | ✓ | ✗ | 12.84 | 1.50 | 27.64 | 0.941 | 0.040 |
| Fused-Planes (RGB) | ✗ | ✓ | ✓ | 63.52 | 0.48 | 27.71 | 0.942 | 0.044 |
| Tri-Planes | ✗ | ✓ | ✗ | 64.08 | 1.50 | 28.56 | 0.953 | 0.035 |
| Fused-Planes-ULW | ✓ | ✗ | ✓ | 7.16 | 0.0008 | 27.51 | 0.935 | 0.063 |
| Fused-Planes | ✓ | ✓ | ✓ | 8.92 | 0.48 | 28.64 | 0.950 | 0.037 |

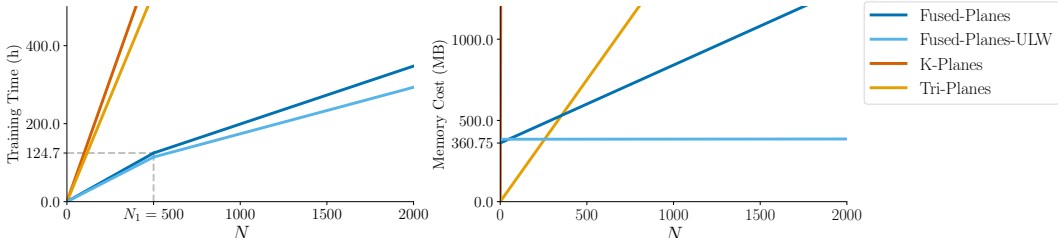

Figure 4: **Scaling the number of objects using planar methods.** Evolution of the total training time (left) and total memory footprint (right) when scaling the number of objects ($N$). As K-Planes is barely visible (right), we present in Figure 6 a magnified version of the memory cost plot.

Importantly, this effect is small, and the resulting quality remains on par with or above that of Tri-Planes. Note that training speed and memory usage remain unchanged in multi-class training with respect to the single-category models.

### 4.3 ADDITIONAL RESULTS

In the Appendix, Figure 10 illustrates a subset of our large-scale results for completeness. Section D.3 provides an ablation study on the number of base planes, in which we show that $M = 50$ is the most effective option for Fused-Planes. Section D.4 presents a rendering speed analysis showing that Fused-Planes achieves substantially faster rendering than Tri-Planes, while being competitive with the other baselines. Section D.5 provides experiments utilizing a low-budget VAE with Fused-Planes, showing that our method exhibits low-sensitivity to the specific VAE initialization. Section D.6 presents a comparison of the total resource costs across our baselines, which shows that Fused-Planes presents competetitve training times, and is the fastest planar method. In terms of memory, Fused-Planes and Fused-Planes-ULW are the most lightweight methods. Section E analyses our base-planes and the representations they learn. In brief, our base planes can be grouped into two categories: semantic planes that clearly encode object-level structures, and residual planes that capture finer intra-class variability. Moreover, we visualize the values of the weights $W_i$ for two different objects, which shows that for each object, a few base planes are dominantly activated, while other planes contribute minor adjustments. Finally, we also present an experiment in which we interpolate between two learned weights in Figure 9, showing that we can transition smoothly from one scene to another. Per-object NVS results and visualizations are available in Tables 17 to 21 and Figures 11 to 15.

### 4.4 ABLATIONS

To justify our design choices, we present an ablation study of our method (Table 5). "**Fused-Planes ($M = 1$)**" reduces the shared base planes $\mathcal{B}$ to a single plane. It demonstrates a slight degradation of quality compared to Fused-Planes, highlighting the necessity for *a set* of base planes. "**Fused-Planes (Micro)**" eliminates the Macro component of Fused-Planes (i.e. $F^{\text{mac}} = 0$), and therefore the shared components. It exhibits lower quality compared to Tri-Planes, which is in line with the degradations seen in Schnepf et al. (2025) for latent NeRFs. In contrast, our full model avoids such issues, underscoring the benefits of shared representations within the latent space, both in quality and

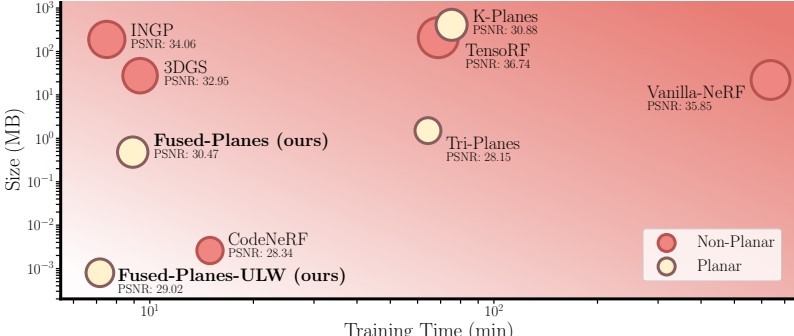

Figure 5: **Resource costs overview.** To reconstruct a large class of objects, one would consider three options: many per-scene models (e.g. INGP, 3DGS, or planar methods), a multi-scene method (e.g. CodeNeRF), or Fused-Planes. Our method presents the lowest per-object training time and memory footprint among all planar representations, while maintaining a similar rendering quality. Circle sizes represent the NVS quality.

memory efficiency. "**Fused-Planes (RGB)**" ablates the latent space and trains Fused-Planes in RGB space. It exhibits lower quality compared to Tri-Planes, and to our full model. Therefore, it shows the necessity of the latent space for making shared representations work effectively. It also highlights the speed improvements enabled by the latent space. "**Tri-Planes**" is equivalent to ablating both the latent space and macro planes, which presents significantly higher resource costs and similar quality. **In summary,** our ablations show that both the latent space and shared representations are needed *concurrently* to avoid quality degradations and minimize resource costs.

### 4.5 LIMITATIONS

Tri-Planes are well-suited for object-centric scenes. However, they exhibit limitations in capturing fine details and handling unbounded scenes, which are characteristic of real-world environments. As such, Tri-Planes cannot be used to reconstruct scenes such as the ones used in the NeRF paper (Mildenhall et al., 2020) or in the Mip-NeRF 360 dataset (Barron et al., 2022). More precisely, to capture fine details, one would need to greatly increase the resolution of each of the Tri-Planes feature grids, leading to significant increases in memory footprint and computation, which undermines the compactness that makes Tri-Planes attractive. Moreover, Tri-Planes assume that the scene fits in a bounded volume, which complicates the modeling of distant backgrounds often present in real scenes. Some methods (Wu et al., 2024b; Lee et al., 2024; Yan et al., 2024) sidestep these limitations by using tricks like utilizing multiple Tri-Planes for large scenes or by modeling only density and relying on other tools for textures. These approaches are beyond this paper's scope.

Since our method adopts the same architecture as Tri-Planes, it also inherits their limitations. Even so, Tri-Planes have been widely adopted (Section 2), as their planar design provides practical advantages despite these drawbacks. Our contribution advances this line of work by proposing a more efficient way to train planar methods at large-scales, while improving the quality of Tri-Planes.

### 5 CONCLUSION

In this work, we introduced Fused-Planes, a novel planar object representation that advances the state of the art in resource-efficient planar 3D modeling and reconstruction of large object classes. This is achieved by shifting away from the traditional approach of reconstructing each object in isolation, and instead exploiting the shared structural similarities within object classes using shared base representations in a specially designed latent space. We showed that Fused-Planes significantly reduces required resources compared to current planar representation, while maintaining rendering quality. Given the recurrent challenges associated with training large-scale planar scene representations, we hope that our contribution will facilitate this task, and make research in image-based models for 3D applications more accessible.

## REPRODUCIBILITY STATEMENT

We have taken several measures to ensure the reproducibility of our findings. The paper includes the necessary implementation details and hyperparameter settings in order to reproduce our results. Additionally, the complete open-source code can be accessed via our project webpage. Together, these resources should allow researchers to fully reproduce and extend our findings.

## ACKNOWLEDGMENTS

We would like to express our sincere gratitude to Thibaut Issenhuth and Song Duong for their thoughtful feedback and insightful comments on this paper. Their careful reading and constructive suggestions helped us clarify key arguments and significantly improve the overall quality of the paper. We are also deeply thankful to Vicky Kalogeiton and Xi Wang for their valuable external feedback. Their fresh perspective encouraged us to rethink several aspects of our work and strengthened the paper in meaningful ways.

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

# A WORKS UTILIZING TRI-PLANES FOR TARGETED TASKS

In this section, we highlight representative works that utilize Tri-Planes for varied targeted tasks.

**Editing.** Bilecen et al. (2025); Ki et al. (2025) use Tri-Planes to perform 3D-aware editing, based on conditioning images. Such editing allows to combine the overall appearance of one object with selected characteristics of a different object.

**Feed-forward reconstruction.** Hong et al. (2024); Wang et al. (2023); Sun et al. (2024) propose feed-forward image-to-3D pipelines: they infer Tri-Planes from single images by switching the output modality of image-based models to Tri-Planes.

**Generation.** Shue et al. (2023); Chen et al. (2023); Anciukevičius et al. (2023) build a diffusion framework around Tri-Planes, treating them as if they were images with more channels, which enables 3D object generation using image generative models.

**Classification.** Cardace et al. (2024) leverage Tri-Planes to classify neural fields without re-creating the explicit signal (i.e. without rendering), and highlight the rich semantic signal present in Tri-Planes, as well as their ease of use with standard neural architectures.

# B ADDITIONAL IMPLEMENTATION DETAILS

This section presents some additional details regarding the training of Fused-Planes, namely its warm-up stage and the early stopping of the encoder.

In practice, we use two regimes of optimization to gain some computational efficiency. In fact, we notice that the encoder $E_\phi$ converges before the set of $N = 2000$ Fused-Planes. Hence, continuing to optimize it would be unnecessary. As such, we jointly train the encoder only with a subset $\mathcal{T}_1 = \{T_1, ..., T_{N_1}\}$ of Fused-Planes (regime 1), before learning the remaining Fused-Planes $\mathcal{T}_2 = \{T_{N_1+1}, ...T_N\}$ with a frozen encoder (regime 2). We set $N_1 = 500$. For completeness, we also detail the warm-up stage of the Fused-Planes (at the start of regimes 1 and 2). This warm-up stage is necessary just after the random initialization of Fused-Planes, to avoid back-propagating random gradients into the auto-encoder.

**Regime 1.** We start by warming-up $\mathcal{T}_1$ with the following objective:

$$\min_{\mathcal{T}_1, \alpha} \sum_{i=1}^{N_1} \sum_{j=1}^{V} \mathcal{L}_{i,j}^{(\text{latent})}(\phi, T_i, \alpha) . \tag{8}$$

We then optimize the Fused-Planes in $\mathcal{T}_1$, the encoder $E_\phi$ and the decoder $D_\psi$ using Equation (7), recalled here:

$$\min_{\mathcal{T}_1, \alpha, \phi, \psi} \sum_{i=1}^{N_1} \sum_{j=1}^{V} \lambda^{(\text{latent})} \mathcal{L}_{i,j}^{(\text{latent})}(\phi, T_i, \alpha)$$
$$+ \lambda^{(\text{RGB})} \mathcal{L}_{i,j}^{(\text{RGB})}(\psi, T_i, \alpha)$$
$$+ \lambda^{(\text{ae})} \mathcal{L}_{i,j}^{(\text{ae})}(\phi, \psi) . \tag{9}$$

**Regime 2.** Similarly to the first regime, we start by warming-up $\mathcal{T}_2$ with the following objective:

$$\min_{\mathcal{T}_2, \alpha} \sum_{i=N_1+1}^{N} \sum_{j=1}^{V} \mathcal{L}_{i,j}^{(\text{latent})}(\phi, T_i, \alpha) . \tag{10}$$

We then optimize the Fused-Planes in $\mathcal{T}_2$, but only $\mathcal{L}^{(\text{RGB})}$ is needed, as the encoder no longer requires training. We keep fine-tuning the decoder $D_\psi$. The objective is:

$$\min_{\mathcal{T}_2, \alpha, \psi} \sum_{i=N_1+1}^{N} \sum_{j=1}^{V} \lambda^{(\text{RGB})} \mathcal{L}_{i,j}^{(\text{RGB})}(\psi, T_i, \alpha) . \tag{11}$$

Practically, we achieve the previous objective using mini-batch gradient descent. Details can be found in Algorithm 1. The rendering quality remains the same between the two regimes, as illustrated in Tables 6 and 7.

# C    DATASET DETAILS

We use ShapeNet (Chang et al., 2015) and Basel-Face (Paysan et al., 2009) to evaluate the novel view synthesis performance of the object representations.

The ShapeNet dataset is a large-scale, annotated collection of 3D models covering various object categories, widely used for 3D applications. We use four distinct object categories to evaluate our method: Cars, Furniture, Speakers and Sofas. For each ShapeNet object, we render $V = 160$ views, sampled from the upper hemisphere surrounding the object.

The Basel-Face dataset contains more than 1000 distinct face models. The faces are generated from a 3D morphable face model with 199 principle components. For faces, we take $V = 50$ front-facing views.

All views are rendered at a resolution of $128 \times 128$. In all our experiments, we use $90\%$ of the views for training and $10\%$ for evaluation.

**Multi-class datasets.** To assess our method in multi-class settings, we construct four new datasets that combine two, three, and four object categories from our original collection. Each dataset contains 2,000 objects. The first dataset, *Speakers & Sofas*, includes 1,000 speakers and 1,000 sofas. The second dataset, *Speakers, Sofas & Furniture*, is composed of 667 speakers, 667 sofas, and 667 furniture objects. The final dataset, *Speakers, Sofas, Furniture & Cars*, contains 500 objects from each of the four categories. We ensure that the scenes used for evaluations are the same across datasets, in order to have rigorous comparisons.

# D    SUPPLEMENTARY RESULTS

## D.1    QUALITATIVE RESULTS

We showcase a subset of our large-scale results on ShapeNet cars in Figure 10.

Additionally, we present in Figures 11 to 15 additional qualitative comparisons across all the methods discussed in our experiments (Section 4). Fused-Plane demonstrates similar visual quality to state-of-the-art methods.

## D.2    QUANTITATIVE RESULTS

Regarding rendering quality, we present per-scene NVS metrics in Tables 17 to 21.

Regarding resource costs, the shared components (i.e. encoder, decoder and base planes) of Fused-Planes and Fused-Planes-ULW respectively require a total of 360.75 MB and 384.19 MB of storage capacity. Note that we do not include the memory footprint of these components in our analysis, as this overhead is constant regardless of the number of objects, and hence negligible in large-scale settings. This memory cost is illustrated in Figure 4 and magnified in Figure 6, focusing on the range [0,100].

Table 6: **Quantitative comparison.** NVS performances on ShapeNet Cars in both regimes of our training.

| | ShapeNet Cars | | | | | |
| | Regime 1 | | | Regime 2 | | |
| | PSNR↑ | SSIM↑ | LPIPS↓ | PSNR↑ | SSIM↑ | LPIPS↓ |
|---|---|---|---|---|---|---|
| Tri-Planes (RGB) | 28.49 | 0.9539 | 0.0291 | 28.58 | 0.9505 | 0.0360 |
| Fused-Planes | 28.14 | 0.9505 | 0.0301 | 28.77 | 0.9496 | 0.0383 |

Table 7: **Quantitative comparison.** NVS performances on Basel Faces in both regimes of our training.

| | Basel Faces | | | | | |
| | Regime 1 | | | Regime 2 | | |
| | PSNR↑ | SSIM↑ | LPIPS↓ | PSNR↑ | SSIM↑ | LPIPS↓ |
|---|---|---|---|---|---|---|
| Tri-Planes (RGB) | 36.82 | 0.9807 | 0.0122 | 36.35 | 0.9787 | 0.0129 |
| Fused-Planes | 36.17 | 0.9678 | 0.0062 | 36.99 | 0.9712 | 0.0056 |

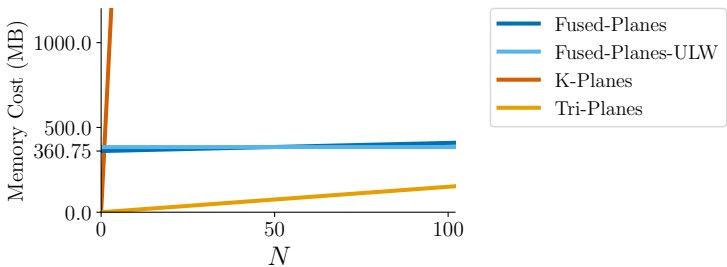

Figure 6: **Memory costs.** This figure presents the memory costs depicted in Figure 4 within the range $N \in [0, 100]$.

### D.3 ABLATION STUDY ON THE NUMBER OF BASE PLANES

Table 8 presents a study of the effect of the number of base planes $M$ on the resource costs and rendering quality of Fused-Planes. The reported NVS quality metrics are averaged on ShapeNet Cars scenes. The table shows that rendering quality varies only minimally across different values of $M$. We select $M = 50$ because it offers the best quality while maintaining similar training time and memory usage. Table 9 presents an extension of this study in the context of datasets mixing multiples objet classes. The results show that Fused-Planes shows similar performances when increasing the number of classes, indicating that class diversity can be effectively captured with 50 base planes, as increasing M beyond 50 does not provide additional benefit.

### D.4 ANALYSIS OF RENDERING SPEED

Table 10 presents the computational costs of computing one Fused-Planes (Equations (1) and (2)) and compares it relative to rendering times. As this operation only needs to be done once when loading a Fused-Planes representation, it is largely negligeable compared to the overall rendering time.

Table 11 presents a comparative study of the rendering speed of our method and its baselines, in terms of frames per second (FPS). Fused-Planes achieve significantly better rendering speeds compared to its planar baselines.

Table 8: **Study on the number of base planes** $M$**.** We select $M = 50$ as it offers the best quality while maintaining similar training time and memory usage.

|  | M | Per-Scene Training Time (min) | Total Memory for 2000 Scenes (MB) | PSNR |
|---|---|---|---|---|
| Fused-Planes | 5 | 8.60 | 1276 | 29.89 |
|  | 20 | 8.61 | 1291 | 30.02 |
|  | 50 | 8.92 | 1322 | 30.27 |
|  | 75 | 8.99 | 1348 | 29.62 |

Table 9: **Effect of M on multi-class reconstruction.** We report PSNR values for NVS evaluation, averaged over multiple scenes of the same category (cars and speakers), when jointly learning one or four classes. Increasing M beyond 50 does not provide additional benefit, indicating that class diversity can be effectively captured with $M = 50$ base planes.

|  | M | One class | | Four classes | |
|---|---|---|---|---|---|
|  |  | Cars | Speakers | Cars | Speakers |
| Fused-Planes | 50 | **30.27** | **29.99** | **29.15** | **29.72** |
|  | 75 | 29.62 | 29.46 | 28.58 | 29.20 |

Note that in both tables, the reported rendering time for Fused-Planes includes (i) the volume-rendering step and (ii) the decoding step that converts the rendered latent representation into an RGB image.

Table 10: **Computational overhead & rendering speed analysis.** We compare the computational cost of Fused-Planes against RGB Tri-Planes for rendering a single frame and multiple frames. Rendering with Fused-Planes is approximately twice as fast. Moreover, the overhead introduced by the Fused-Planes computation (Equations (1) and (2)), which only needs to be done once, is largely negligible compared to the overall rendering time.

|  | Compute Fused-Planes (Eq. 1 & 2; $\downarrow$) | Render 1 frame ($\downarrow$) | Render 30s video @ 30 fps ($\downarrow$) |
|---|---|---|---|
| Tri-Planes | — | 23.30 ms | 20.97 s |
| Fused-Planes | 0.65 ms | **10.95 ms** | **9.85 s** |

Table 11: **Rendering speed comparison (FPS).** We compare the rendering FPS of our method with the baselines during inference. The Nerfstudio implementations are used for all baseline models. Fused-Planes showcases significantly faster rendering speed than all baselines except 3DGS.

|  | FPS ($\uparrow$) |
|---|---|
| Vanilla-NeRF | 0.85 |
| Instant-NGP | 48.7 |
| TensoRF | 13.6 |
| 3DGS | **176.0** |
| K-Planes | 14.3 |
| Tri-Planes | 42.9 |
| Fused-Planes | 91.3 |

## D.5 RESULTS USING A LOW-BUDGET VAE

Table 12 reports results where we train Fused-Planes with a VAE that has been reset (all weights are randomly initialized) and trained on our scenes with a low budget. In order to avoid backpropagating random gradients to the modules in Fused-Planes at initialization, we allocate $15\%$ of training time to warm up the VAE after its reset, using the images of the scenes. This is necessary as training Fused-Planes with a non-functional VAE makes it impossible for Fused-Planes to learn the scenes.

This experiment is conducted on 2000 scenes from ShapeNet Cars. The results indicate that employing a VAE trained on a smaller dataset and with lower budget introduces only minor degradation in output quality. This suggests that our framework exhibits low sensitivity to the specific initialization of the VAE.

Table 12: **Results using a low-budget VAE.** Using a low-budget VAE with Fused-Planes leads to only minor quality degradation, showing that our mehthod is robust to VAE initialization.

|  | PSNR | SSIM | LPIPS |
|---|---|---|---|
| Fused-Planes (low-budget VAE) | 29.22 | 0.953 | 0.035 |
| Fused-Planes | 30.27 | 0.960 | 0.033 |

## D.6 TOTAL COST COMPARISON ACROSS DIFFERENT VALUES OF N

For completeness, we report total training time (Table 13) and total memory footprint (Table 14) when varying the number of object $N$ being learned. The results show that Fused-Planes presents competitive training times, and is the fastest planar method. In terms of memory, Fused-Planes and Fused-Planes-ULW are the most lightweight methods

Table 13: **Total training time across different values of N.** Fused-Planes is the fastest planar method, and present competitive training times compared to other non-planar baselines. All training times are reported in days.

|  | Planar | Total training time (days) | | | | |
|---|---|---|---|---|---|---|
|  |  | $N = 1000$ | $N = 2000$ | $N = 5000$ | $N = 10000$ | $N = 20000$ |
| Vanilla-NeRF | ✗ | 442.2 | 884.4 | 2211.1 | 4422.2 | 8844.4 |
| Instant-NGP | ✗ | 5.2 | 10.4 | 26.1 | 52.2 | 104.4 |
| TensoRF | ✗ | 47.9 | 95.7 | 239.3 | 478.7 | 957.4 |
| 3DGS | ✗ | 6.5 | 13.0 | 32.5 | 65.1 | 130.1 |
| K-Planes | ✓ | 52.3 | 104.7 | 261.6 | 523.3 | 1046.5 |
| Tri-Planes | ✓ | 44.7 | 89.3 | 223.3 | 446.7 | 893.3 |
| Fused-Planes-ULW | ✓ | 7.2 | 12.2 | 27.1 | 52.0 | 101.7 |
| Fused-Planes | ✓ | 8.3 | 14.5 | 33.2 | 64.3 | 126.5 |

## D.7 MEMORY BREAKDOWN

Table 15 provides a breakdown of the memory footprint of the different components used. The memory cost required by a single object is notably low compared to our baselines in the main paper. The memory cost required by our shared components can be considered as an acceptable entry cost, as its value is less than a single K-Planes representation.

# E BASE PLANES ANALYSIS

To further investigate the learned representations in our base planes, we visualize their contents, analyze the values in the weights $W_i$, and interpolate between different weights. We present our analysis below.

Table 14: **Total memory cost across different values of N.** Fused-Planes is the most lightweight method among all baselines. Sizes are reported in GB.

| | Planar | Total memory footprint (GB) | | | | |
|---|---|---|---|---|---|---|
| | | $N = 1000$ | $N = 2000$ | $N = 5000$ | $N = 10000$ | $N = 20000$ |
| Vanilla-NeRF | ✗ | 21.5 | 43.0 | 107.4 | 214.8 | 429.7 |
| Instant-NGP | ✗ | 184.7 | 369.4 | 923.5 | 1847.0 | 3694.0 |
| TensoRF | ✗ | 203.4 | 406.9 | 1017.2 | 2034.4 | 4068.7 |
| 3DGS | ✗ | 27.0 | 54.0 | 135.1 | 270.1 | 540.2 |
| K-Planes | ✓ | 400.6 | 801.1 | 2002.8 | 4005.6 | 8011.1 |
| Tri-Planes | ✓ | 1.5 | 2.9 | 7.3 | 14.6 | 29.3 |
| Fused-Planes-ULW | ✓ | 0.4 | 0.4 | 0.4 | 0.4 | 0.4 |
| Fused-Planes | ✓ | 0.8 | 1.3 | 2.7 | 5.0 | 9.7 |

Table 15: **Memory breakdown.** This table breaks down the memory footprints of the different components in our pipeline. Note that the memory usage of shared components remains constant and does not depend on the number of objects. In contrast, the memory footprint for storing objects *increases linearly* with the number of objects. Therefore, in large-scale settings, the dominant factor is the memory that *increases* with the number of objects, as illustrated in Figure 4.

| Module | Shared? | Size |
|---|---|---|
| Encoder $E_\phi$ | ✓ | 130.38 MB |
| Decoder $D_\psi$ | ✓ | 178.86 MB |
| $50\times$ base planes $\mathcal{B}$ ($F^{\text{mac}} = 22$) | ✓ | 51.5 MB |
| $1\times$ tiny MLP (renderer $R_\alpha$) | ✓ | 14.27 KB |
| $1\times$ micro plane $T_i^{\text{mic}}$ ($F^{\text{mic}} = 10$) | ✗ | 480 KB |
| $1\times$ weight $W_i$ | ✗ | 811 B |
| Memory footprint of shared components | ✓ | 360.75 MB |
| Memory footprint of a single object | ✗ | 0.481 MB |

**Protocol for base planes visualizations.** Recall that, in our standard pipeline, we render a learned Fused-Planes-ULW representation $T_i$ (corresponding to scene $i$) using volume rendering followed by a decoder, where each fused representation is defined as:

$$T_i = \sum_{k=1}^{M} w_i^k B_k \,, \tag{12}$$

where $T_i$ is the ultra-lightweight variant of our method (i.e. no micro planes).

To visualize our base planes, we do not render $T_i$. Instead, we directly render individual base planes $B_k$, using our Fused-Planes-ULW model trained on ShapeNet Cars and Basel Faces datasets. This is indeed possible for our Fused-Planes-ULW model, as each $B_k$ has the same dimensionality as $T_i$.

Using this protocol, we visualize 10 different base planes in Figure 7. As illustrated in the figure, the base planes can be grouped into two categories: (i) semantic: some base planes clearly encode object-level structures (e.g. faces, cars), (ii) residual: other base planes capture finer intra-class variability relative to the semantic base planes. Together, these base planes contribute to each object representation.

Moreover, we visualize the values of the weights $W_i$ for two different cars. The results are presented in Figure 8. We observe that a few base planes dominate the final fused representation, and the dominant planes vary across scenes, while other base planes contribute only minor adjustments.

Finally, we illustrate in Figure 9 the Fused-Planes-ULW resulting from a weight interpolation. Specifically, we first choose two weights $W_1$ and $W_2$ corresponding to two scenes. We then compute

$W_t = tW_1 + (1 - t)W_2$ for $t \in \{0, 0.25, 0.5, 0.75, 1\}$. Injecting $W_t$ into Equation (12) yields a set of Fused-Planes-ULW, which we render and visualize.

We observe that interpolating between weights yield coherent scenes, where we transition smoothly from one scene to another (e.g. the mouth closes gradually across the different faces).

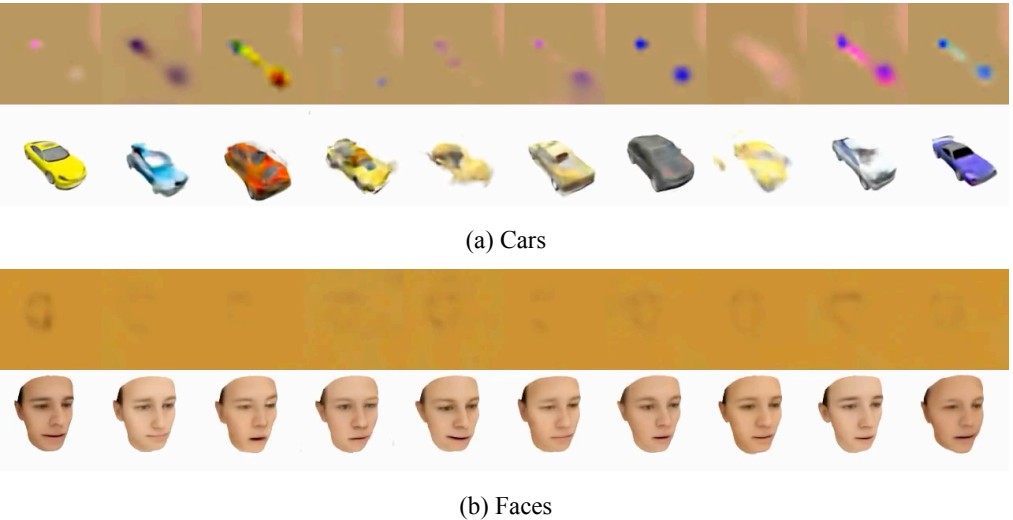

(a) Cars

(b) Faces

Figure 7: **Base planes visualizations.** We observe that some base planes clearly encode object-level structures, while other encode finer intra-class variability. Together, these base planes contribute to each object representation.

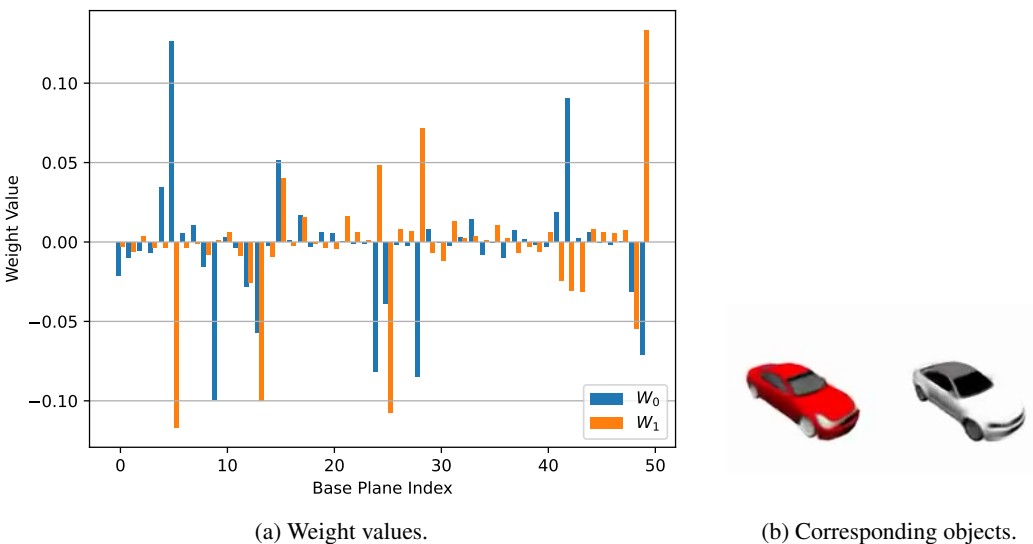

(a) Weight values.

(b) Corresponding objects.

Figure 8: **Learned weights $W_i$ for two scenes of our ULW model.** The weight $W_i \in \mathbb{R}^M$ is learned to linearly decompose a Fused-Planes $T_i$ on the set of base planes $\{B_k\}$ using Equation (12). In this figure, we show the learned weights (left) corresponding to two objects (right). We notice that a few base planes dominate the final fused representation, and the dominant planes vary across scenes, while other base planes only contribute to minor adjustments

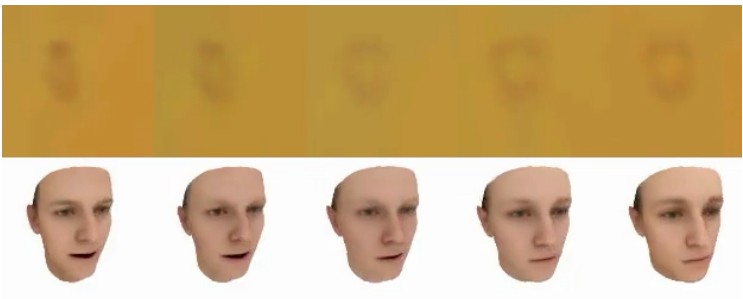

Figure 9: **Interpolation between base planes weights.** We linearly interpolate between the weights of two scenes (leftmost and rightmost). We observe that this interpolation leads to coherent structures, where we transition smoothly from one scene to another.

## F    CODENERF-A

**CodeNeRF.** CodeNeRF (Jang & Agapito, 2021) learns a set of scenes with a single neural representation $f_\theta$ which is conditioned on scene-specific latent codes. Specifically, for each scene, a shape code $z_s$ and an appearance code $z_a$ is learned, such that $f_\theta(z_s, z_a)$ models the current scene. Once the conditional NeRF $f_\theta$ is trained on a large set of scenes, it can learn new scenes using test-time optimization. This test-time optimization consists of learning a new scene by optimizing only the codes $(z_s, z_a)$, while keeping $f_\theta$ fixed. By reducing the number of trainable parameters, test-time-optimization offers increased training speed. Furthermore, the memory required to store an additional scene on disk is very low, since only $(z_s, z_a)$ need to be stored.

**CodeNeRF-A.** In our experiments, we introduce CodeNeRF-A as a new comparative baseline. CodeNeRF-A employs a novel training procedure inspired by ours, which leverages the test-time optimization method originally proposed by CodeNeRF to improve efficiency for learning multiple scenes. Specifically, we first train the shared neural representation $f_\theta$ of CodeNeRF on a subset $O_1$ of $\mathcal{O}$ composed of $N_1$ scenes. Subsequently, we employ test-time-optimization with the previously trained representation to learn the remaining scenes $O_2$, with lowered training times.

We present in Table 16 a comparison of CodeNeRF-A performances when taking $N_1 = 500$ and $N_1 = 1000$. CodeNeRF-A showcases better performances with $N_1 = 1000$, which we set throughout the paper for this method.

Table 16: **Choice of $N_1$ for CodeNeRF-A.** CodeNeRF-A showcases better performances when taking $N_1 = 1000$, which we set throughout the paper for this method.

|  | $N_1$ | ShapeNet datasets | | | Basel Faces | | |
| --- | --- | --- | --- | --- | --- | --- | --- |
|  |  | PSNR | SSIM | LPIPS | PSNR | SSIM | LPIPS |
| CodeNeRF-A | 500 | 26.81 | 0.9108 | 0.1281 | 34.15 | 0.964 | 0.011 |
| CodeNeRF-A | 1000 | 26.99 | 0.9154 | 0.1257 | 35.44 | 0.971 | 0.010 |

## G    HYPERPARAMETERS

For reproducibility purposes, Tables 22 and 23 expose our hyperparameter settings respectively for the first and second regimes of our training. A more detailed list of our hyperparameters can be found in the configuration files of our open-source code.

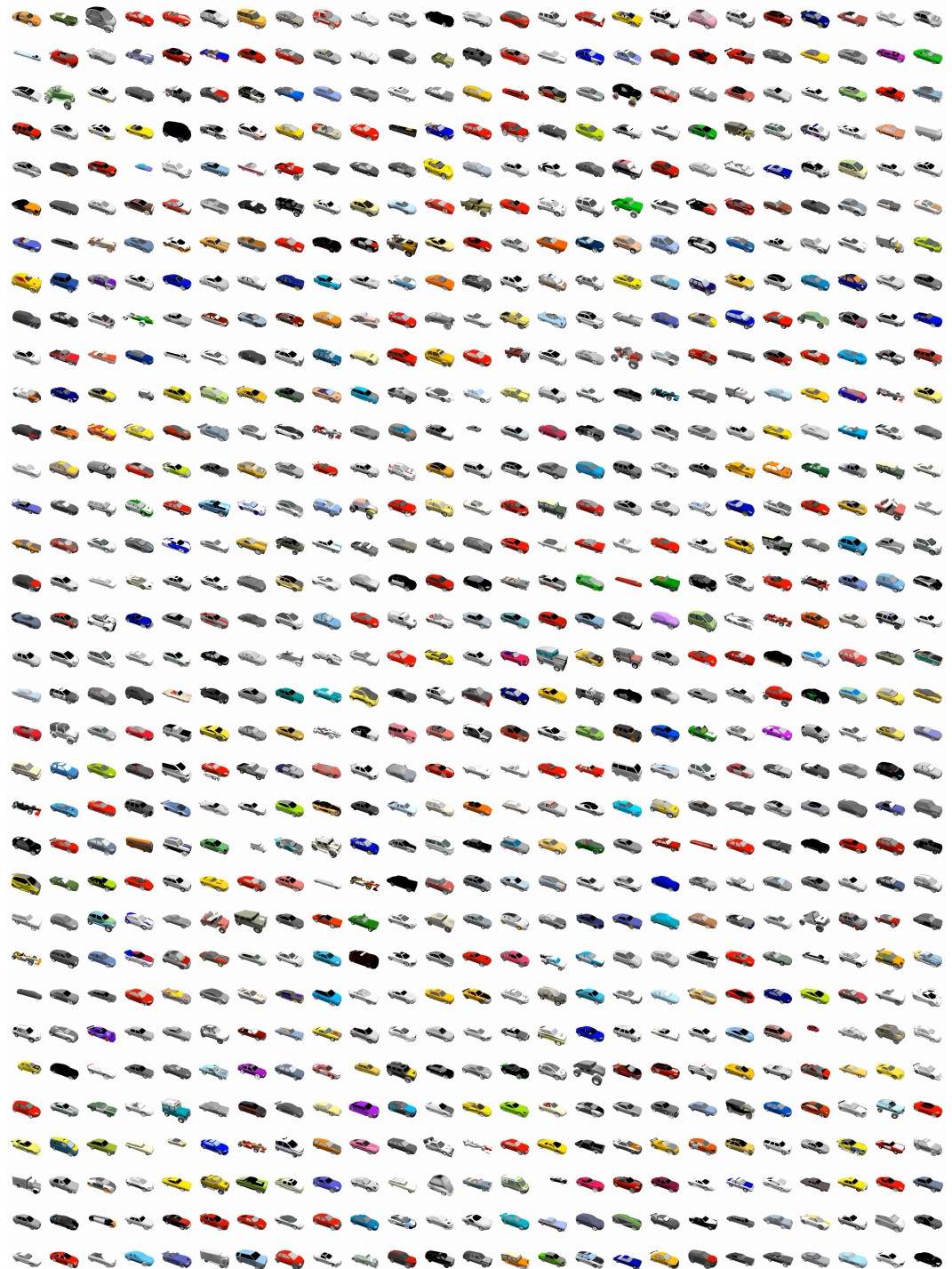

Figure 10: **Large-scale results.** We qualitatively show a subset of our large-scale results on ShapeNet cars.

Table 17: **Per-object quantitative comparison on Basel Faces.**

| | Planar | Face 1 | | | Face 2 | | | Face 3 | | | Face 4 | | |
|---|---|---|---|---|---|---|---|---|---|---|---|---|---|
| | | PSNR | SSIM | LPIPS | PSNR | SSIM | LPIPS | PSNR | SSIM | LPIPS | PSNR | SSIM | LPIPS |
| Vanilla-NeRF | ✗ | 43.44 | 0.996 | 0.001 | 43.90 | 0.997 | 0.001 | 42.65 | 0.996 | 0.001 | 41.64 | 0.994 | 0.003 |
| Instant-NGP | ✗ | 37.79 | 0.987 | 0.004 | 40.01 | 0.990 | 0.002 | 35.38 | 0.977 | 0.013 | 32.96 | 0.969 | 0.016 |
| TensoRF | ✗ | 40.80 | 0.993 | 0.003 | 42.72 | 0.995 | 0.001 | 40.96 | 0.993 | 0.003 | 38.16 | 0.982 | 0.011 |
| 3DGS | ✗ | 43.69 | 0.998 | 0.001 | 45.41 | 0.998 | 0.001 | 43.22 | 0.997 | 0.001 | 39.93 | 0.986 | 0.007 |
| CodeNeRF | ✗ | 35.49 | 0.974 | 0.009 | 36.35 | 0.974 | 0.006 | 34.42 | 0.970 | 0.012 | 35.60 | 0.971 | 0.012 |
| CodeNeRF-A | ✗ | 36.25 | 0.974 | 0.008 | 37.14 | 0.977 | 0.005 | 32.49 | 0.961 | 0.015 | 35.87 | 0.972 | 0.013 |
| K-Planes | ✓ | 40.68 | 0.993 | 0.003 | 39.46 | 0.988 | 0.010 | 41.11 | 0.993 | 0.004 | 39.68 | 0.990 | 0.004 |
| Tri-Planes | ✓ | 36.05 | 0.978 | 0.015 | 37.46 | 0.982 | 0.011 | 36.78 | 0.980 | 0.014 | 35.59 | 0.977 | 0.012 |
| Fused-Planes-ULW | ✓ | 33.84 | 0.950 | 0.013 | 35.00 | 0.958 | 0.007 | 33.41 | 0.959 | 0.011 | 33.58 | 0.954 | 0.010 |
| Fused-Planes | ✓ | 36.24 | 0.970 | 0.007 | 38.63 | 0.975 | 0.004 | 37.04 | 0.975 | 0.006 | 37.04 | 0.971 | 0.006 |

Table 18: **Per-object quantitative comparison on ShapeNet Cars.**

| | Planar | Car 1 | | | Car 2 | | | Car 3 | | | Car 4 | | |
|---|---|---|---|---|---|---|---|---|---|---|---|---|---|
| | | PSNR | SSIM | LPIPS | PSNR | SSIM | LPIPS | PSNR | SSIM | LPIPS | PSNR | SSIM | LPIPS |
| Vanilla-NeRF | ✗ | 38.43 | 0.995 | 0.003 | 41.20 | 0.995 | 0.005 | 37.43 | 0.994 | 0.005 | 39.53 | 0.995 | 0.003 |
| Instant-NGP | ✗ | 35.31 | 0.986 | 0.008 | 37.88 | 0.990 | 0.010 | 34.06 | 0.987 | 0.013 | 36.33 | 0.989 | 0.007 |
| TensoRF | ✗ | 38.66 | 0.994 | 0.003 | 40.55 | 0.995 | 0.005 | 37.80 | 0.995 | 0.004 | 40.00 | 0.995 | 0.003 |
| 3DGS | ✗ | 32.00 | 0.966 | 0.057 | 38.74 | 0.993 | 0.010 | 35.41 | 0.985 | 0.024 | 37.51 | 0.994 | 0.006 |
| CodeNeRF | ✗ | 27.87 | 0.950 | 0.055 | 28.05 | 0.937 | 0.097 | 26.19 | 0.929 | 0.088 | 27.07 | 0.930 | 0.075 |
| CodeNeRF-A | ✗ | 27.10 | 0.946 | 0.055 | 26.86 | 0.929 | 0.103 | 25.25 | 0.921 | 0.092 | 27.10 | 0.932 | 0.074 |
| K-Planes | ✓ | 30.51 | 0.966 | 0.029 | 33.84 | 0.976 | 0.027 | 29.73 | 0.968 | 0.037 | 30.57 | 0.967 | 0.031 |
| Tri-Planes | ✓ | 30.11 | 0.962 | 0.024 | 30.13 | 0.949 | 0.043 | 28.86 | 0.949 | 0.040 | 29.67 | 0.950 | 0.039 |
| Fused-Planes-ULW | ✓ | 27.60 | 0.938 | 0.054 | 29.91 | 0.948 | 0.064 | 28.44 | 0.945 | 0.050 | 28.87 | 0.942 | 0.051 |
| Fused-Planes | ✓ | 30.15 | 0.964 | 0.021 | 31.20 | 0.961 | 0.043 | 29.69 | 0.958 | 0.035 | 30.05 | 0.954 | 0.033 |

Table 19: **Per-object quantitative comparison on ShapeNet Sofas.**

| | Planar | Sofa 1 | | | Sofa 2 | | | Sofa 3 | | | Sofa 4 | | |
|---|---|---|---|---|---|---|---|---|---|---|---|---|---|
| | | PSNR | SSIM | LPIPS | PSNR | SSIM | LPIPS | PSNR | SSIM | LPIPS | PSNR | SSIM | LPIPS |
| Vanilla-NeRF | ✗ | 31.06 | 0.966 | 0.034 | 31.83 | 0.965 | 0.032 | 33.58 | 0.940 | 0.122 | 36.82 | 0.984 | 0.013 |
| Instant-NGP | ✗ | 29.91 | 0.969 | 0.031 | 33.60 | 0.975 | 0.027 | 35.42 | 0.974 | 0.013 | 35.54 | 0.977 | 0.016 |
| TensoRF | ✗ | 32.92 | 0.987 | 0.011 | 37.17 | 0.992 | 0.010 | 37.47 | 0.987 | 0.009 | 37.98 | 0.987 | 0.013 |
| 3DGS | ✗ | 30.85 | 0.986 | 0.020 | 33.95 | 0.989 | 0.025 | 34.60 | 0.982 | 0.023 | 33.46 | 0.984 | 0.047 |
| CodeNeRF | ✗ | 25.47 | 0.938 | 0.113 | 29.80 | 0.938 | 0.068 | 29.18 | 0.919 | 0.139 | 30.14 | 0.944 | 0.100 |
| CodeNeRF-A | ✗ | 24.61 | 0.928 | 0.121 | 29.67 | 0.936 | 0.067 | 28.05 | 0.899 | 0.130 | 29.39 | 0.938 | 0.092 |
| K-Planes | ✓ | 25.84 | 0.947 | 0.054 | 32.28 | 0.974 | 0.028 | 32.90 | 0.968 | 0.028 | 32.59 | 0.964 | 0.037 |
| Tri-Planes | ✓ | 26.34 | 0.929 | 0.082 | 29.24 | 0.930 | 0.091 | 28.89 | 0.903 | 0.121 | 29.43 | 0.922 | 0.118 |
| Fused-Planes-ULW | ✓ | 24.72 | 0.921 | 0.130 | 30.29 | 0.938 | 0.047 | 29.99 | 0.917 | 0.091 | 31.06 | 0.947 | 0.069 |
| Fused-Planes | ✓ | 27.83 | 0.964 | 0.020 | 31.75 | 0.958 | 0.024 | 31.71 | 0.945 | 0.032 | 32.39 | 0.963 | 0.038 |

Table 20: **Per-object quantitative comparison on ShapeNet Speakers.**

| | Planar | Speaker 1 | | | Speaker 2 | | | Speaker 3 | | | Speaker 4 | | |
|---|---|---|---|---|---|---|---|---|---|---|---|---|---|
| | | PSNR | SSIM | LPIPS | PSNR | SSIM | LPIPS | PSNR | SSIM | LPIPS | PSNR | SSIM | LPIPS |
| Vanilla-NeRF | ✗ | 35.95 | 0.962 | 0.065 | 30.97 | 0.980 | 0.021 | 35.41 | 0.970 | 0.032 | 33.65 | 0.977 | 0.015 |
| Instant-NGP | ✗ | 36.56 | 0.983 | 0.016 | 27.31 | 0.955 | 0.043 | 34.75 | 0.980 | 0.024 | 31.16 | 0.966 | 0.022 |
| TensoRF | ✗ | 38.73 | 0.989 | 0.012 | 29.74 | 0.978 | 0.024 | 37.57 | 0.988 | 0.017 | 33.60 | 0.976 | 0.016 |
| 3DGS | ✗ | 34.45 | 0.981 | 0.059 | 24.21 | 0.870 | 0.095 | 31.70 | 0.979 | 0.071 | 28.11 | 0.940 | 0.071 |
| CodeNeRF | ✗ | 29.81 | 0.894 | 0.133 | 24.91 | 0.931 | 0.106 | 28.85 | 0.925 | 0.178 | 28.26 | 0.935 | 0.126 |
| CodeNeRF-A | ✗ | 23.73 | 0.875 | 0.167 | 24.32 | 0.922 | 0.114 | 27.31 | 0.905 | 0.184 | 26.11 | 0.918 | 0.129 |
| K-Planes | ✓ | 33.80 | 0.969 | 0.034 | 21.84 | 0.905 | 0.102 | 32.33 | 0.963 | 0.046 | 27.19 | 0.923 | 0.069 |
| Tri-Planes | ✓ | 29.25 | 0.911 | 0.147 | 23.11 | 0.903 | 0.098 | 29.30 | 0.914 | 0.160 | 26.41 | 0.907 | 0.132 |
| Fused-Planes-ULW | ✓ | 30.38 | 0.932 | 0.134 | 26.75 | 0.948 | 0.049 | 29.93 | 0.940 | 0.104 | 29.83 | 0.942 | 0.063 |
| Fused-Planes | ✓ | 32.89 | 0.966 | 0.042 | 26.40 | 0.949 | 0.046 | 30.63 | 0.951 | 0.066 | 30.04 | 0.947 | 0.057 |

Table 21: **Per-object quantitative comparison on ShapeNet Furnitures.**

| | Planar | Furniture 1 | | | Furniture 2 | | | Furniture 3 | | | Furniture 4 | | |
|---|---|---|---|---|---|---|---|---|---|---|---|---|---|
| | | PSNR | SSIM | LPIPS | PSNR | SSIM | LPIPS | PSNR | SSIM | LPIPS | PSNR | SSIM | LPIPS |
| Vanilla-NeRF | ✗ | 38.42 | 0.976 | 0.014 | 35.56 | 0.985 | 0.015 | 38.01 | 0.978 | 0.018 | 35.75 | 0.965 | 0.028 |
| Instant-NGP | ✗ | 35.12 | 0.951 | 0.032 | 33.64 | 0.977 | 0.023 | 33.91 | 0.954 | 0.035 | 34.49 | 0.959 | 0.031 |
| TensoRF | ✗ | 38.03 | 0.974 | 0.018 | 36.23 | 0.989 | 0.013 | 36.22 | 0.973 | 0.021 | 35.23 | 0.964 | 0.031 |
| 3DGS | ✗ | 34.49 | 0.991 | 0.033 | 31.14 | 0.989 | 0.051 | 33.15 | 0.975 | 0.050 | 33.49 | 0.998 | 0.052 |
| CodeNeRF | ✗ | 29.82 | 0.909 | 0.157 | 27.22 | 0.928 | 0.139 | 30.20 | 0.932 | 0.224 | 30.68 | 0.944 | 0.139 |
| CodeNeRF-A | ✗ | 28.65 | 0.868 | 0.156 | 26.18 | 0.914 | 0.148 | 28.47 | 0.899 | 0.234 | 29.11 | 0.918 | 0.146 |
| K-Planes | ✓ | 34.41 | 0.951 | 0.028 | 30.72 | 0.960 | 0.044 | 33.01 | 0.949 | 0.045 | 32.51 | 0.941 | 0.055 |
| Tri-Planes | ✓ | 26.88 | 0.875 | 0.175 | 27.17 | 0.913 | 0.180 | 28.05 | 0.902 | 0.237 | 27.58 | 0.886 | 0.250 |
| Fused-Planes-ULW | ✓ | 25.80 | 0.891 | 0.219 | 29.91 | 0.947 | 0.073 | 29.67 | 0.938 | 0.173 | 31.20 | 0.958 | 0.102 |
| Fused-Planes | ✓ | 30.19 | 0.962 | 0.030 | 30.54 | 0.957 | 0.039 | 29.81 | 0.947 | 0.102 | 32.34 | 0.972 | 0.042 |

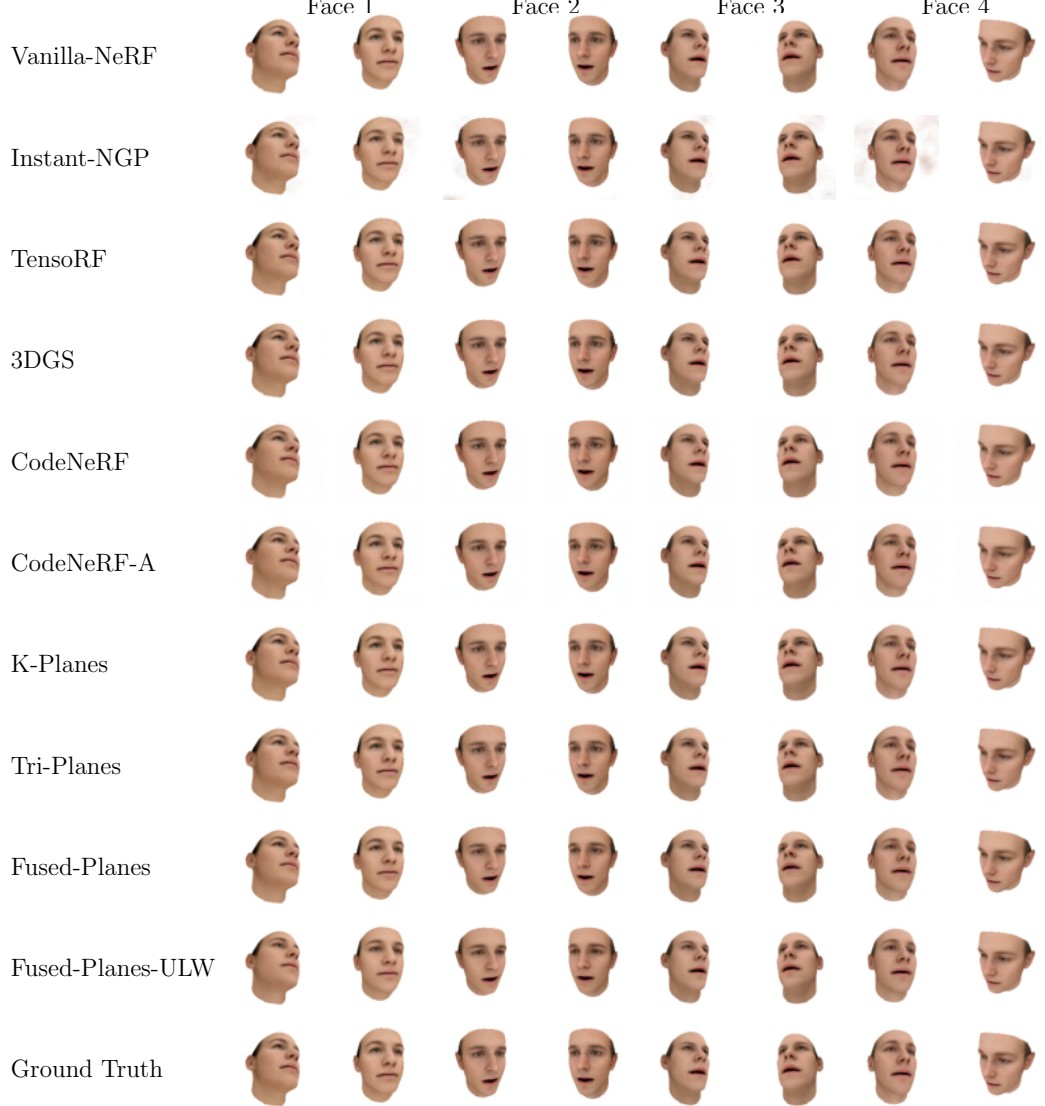

Figure 11: **Qualitative comparison.** Comparison of NVS quality on test views of four objects from Basel Faces.

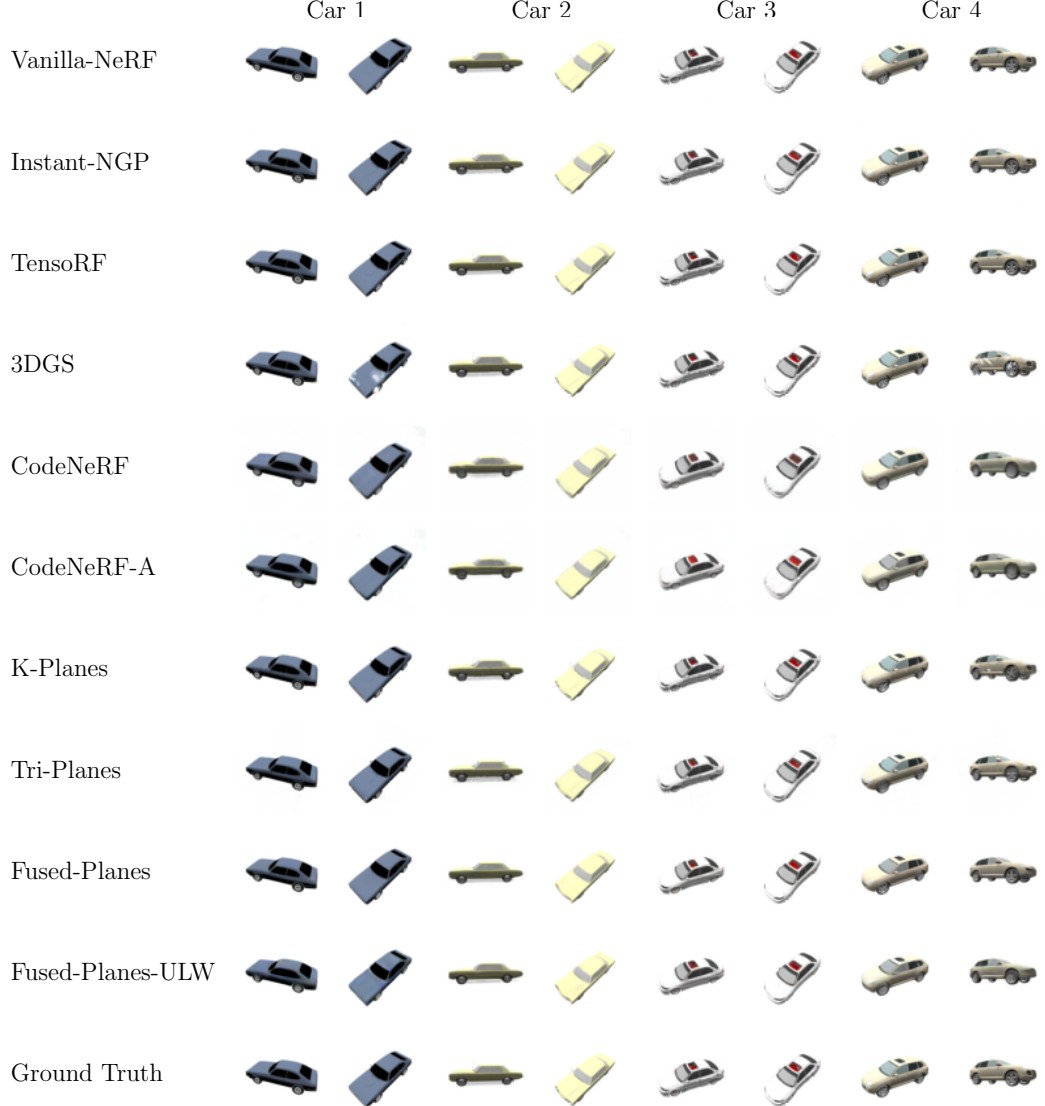

Figure 12: **Qualitative comparison.** Comparison of NVS quality on test views of four objects from the Cars category of ShapeNet.

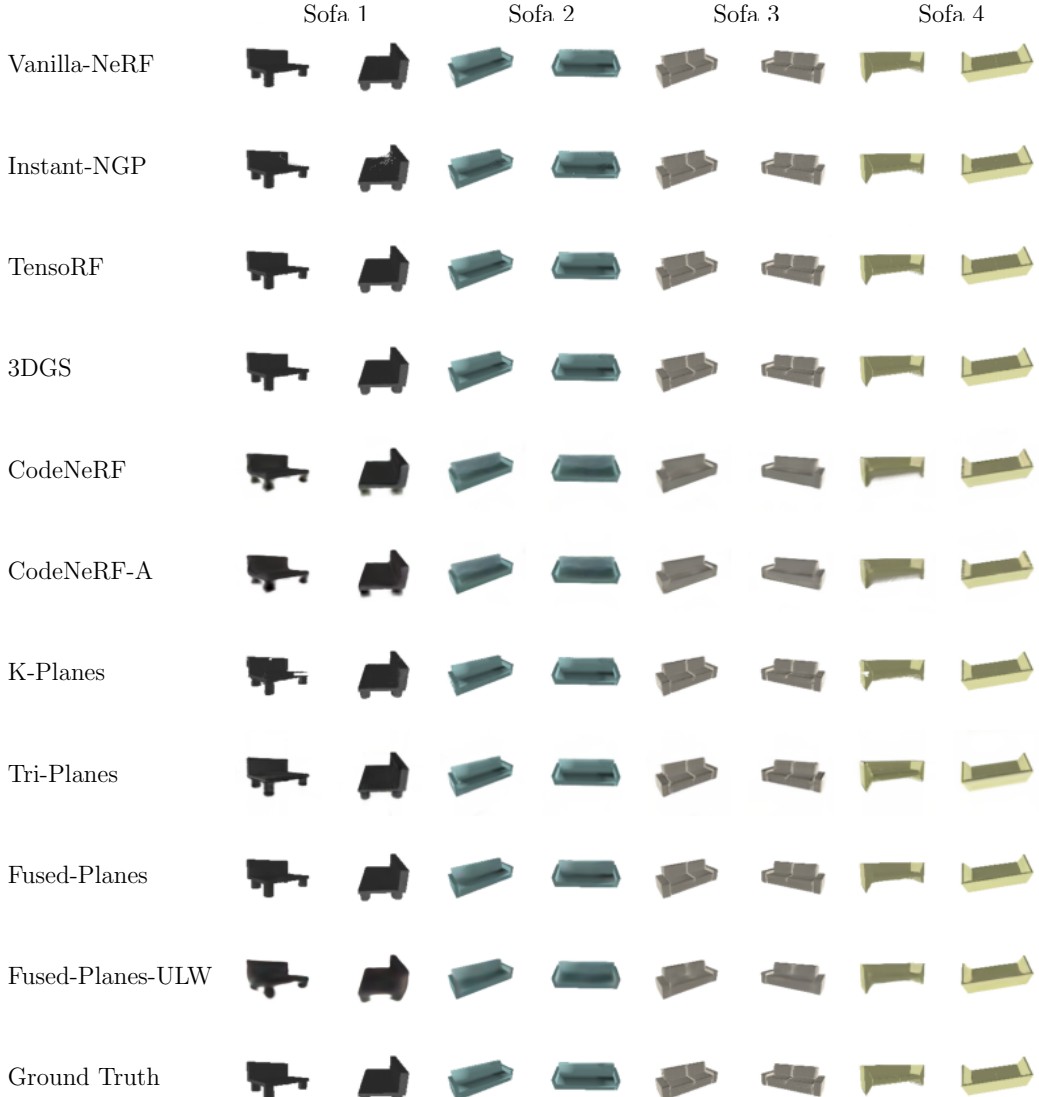

Figure 13: **Qualitative comparison.** Comparison of NVS quality on test views of four objects from the Sofas category of ShapeNet.

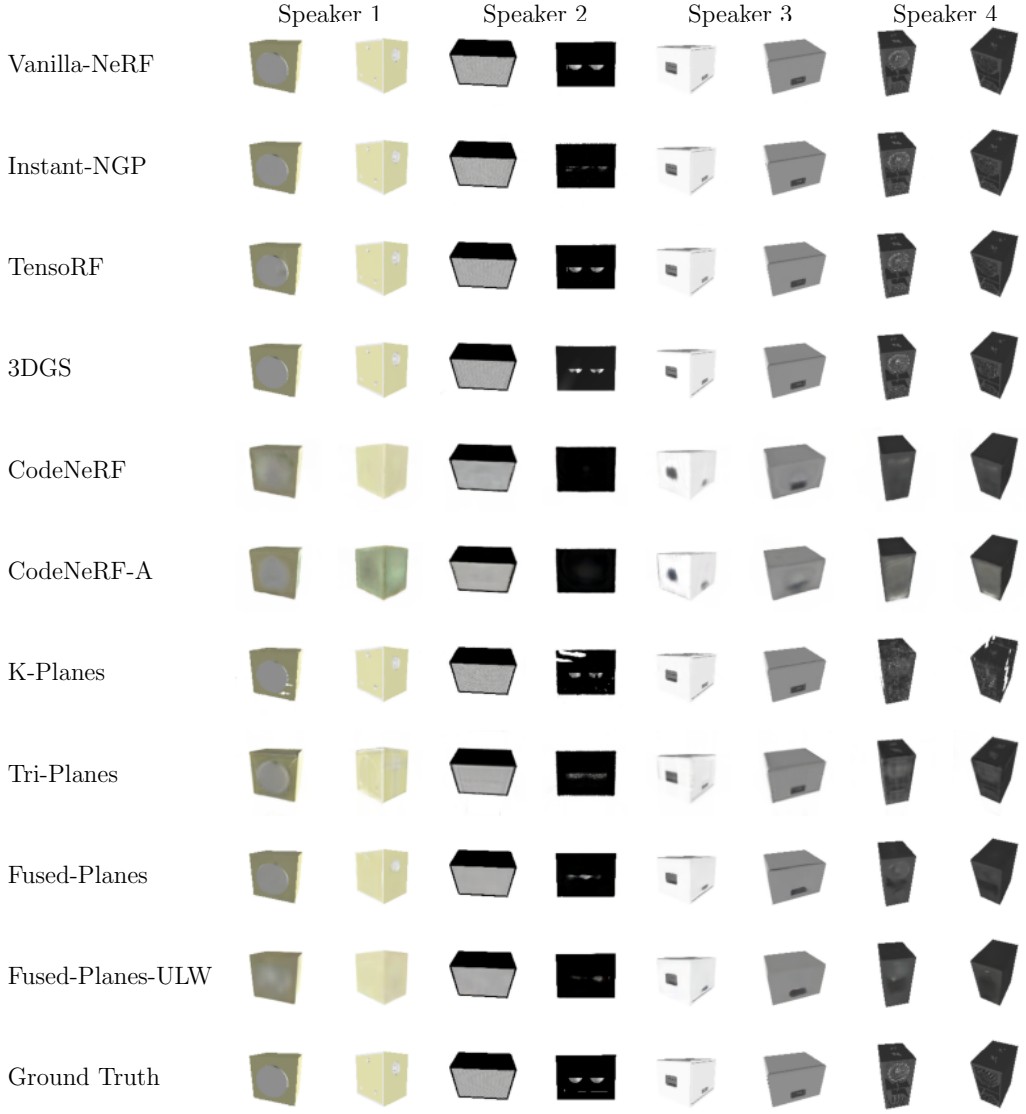

Figure 14: **Qualitative comparison.** Comparison of NVS quality on test views of four objects from the Speakers category of ShapeNet.

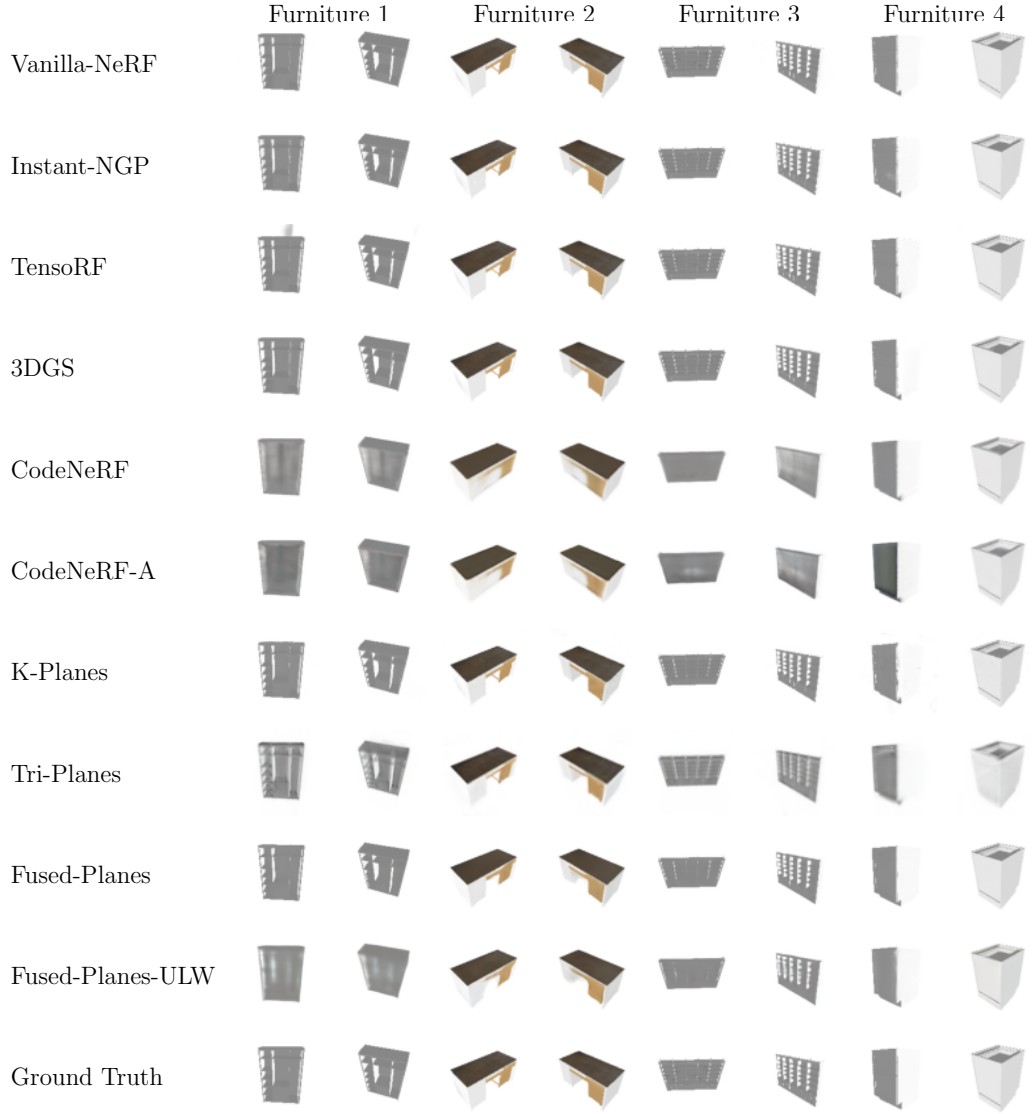

Figure 15: **Qualitative comparison.** Comparison of NVS quality on test views of four objects from the Furniture category of ShapeNet.

we aim to improve upon the resource costs

Table 22: **Fused-Planes regime 1 hyperparameters.**

| Parameter | Value |
|---|---|
| General | |
| Number of scenes $N$ | 2000 |
| Number of scenes for regime 1 $N_1$ | 500 |
| Pretraining epochs | 50 |
| Number of epochs $N_{\mathrm{epoch}}^{(1)}$ | 50 |
| Fused-Planes | |
| Number of micro feature $F_{\mathrm{mic}}$ | 10 |
| Number of macro feature $F_{\mathrm{mac}}$ | 22 |
| Number of base plane $M$ | 50 |
| Tri-Planes resolution | 64 |
| Loss | |
| $\lambda^{(\mathrm{latent})}$ | 1 |
| $\lambda^{(\mathrm{RGB})}$ | 1 |
| $\lambda^{(\mathrm{ae})}$ | 0.1 |
| Optimization (warm-up) | |
| Optimizer | Adam |
| Batch size | 512 |
| Learning rate (Micro planes $T_i^{(\mathrm{mic})}$) | $10^{-2}$ |
| Learning rate (Renderer $R_\alpha$) | $10^{-2}$ |
| Learning rate (Weights $W_i$) | $10^{-2}$ |
| Learning rate (Base planes $B_k$) | $10^{-2}$ |
| Scheduler | Multistep |
| Decay factor | 0.3 |
| Decay milestones | $[20, 40]$ |
| Optimization (training) | |
| Optimizer | Adam |
| Batch size | 32 |
| Learning rate (encoder) | $10^{-4}$ |
| Learning rate (decoder) | $10^{-4}$ |
| Learning rate (Micro planes $T_i^{(\mathrm{mic})}$) | $10^{-4}$ |
| Learning rate (Renderer $R_\alpha$) | $10^{-4}$ |
| Learning rate (Weights $W_i$) | $10^{-2}$ |
| Learning rate (Base planes $B_k$) | $10^{-2}$ |
| Scheduler | Multistep |
| Decay factor | 0.3 |
| Decay milestones | $[20, 40]$ |

Table 23: **Fused-Planes regime 2 hyperparameters.**

| Parameter | Value |
|---|---|
| General | |
| Number of scenes $N$ | 2000 |
| Number of epochs $N_{\mathrm{epoch}}^{(2)}$ | 80 |
| Number of warm-up epochs $N_{\mathrm{epoch}}^{(\mathrm{WU})}$ | 30 |
| Fused-Planes | |
| Number of micro feature $F_{\mathrm{mic}}$ | 10 |
| Number of macro feature $F_{\mathrm{mac}}$ | 22 |
| Number of base plane $M$ | 50 |
| Tri-Planes resolution | 64 |
| Loss | |
| $\lambda^{(\mathrm{latent})}$ | 1 |
| $\lambda^{(\mathrm{RGB})}$ | 1 |
| Optimization (Warm-up) | |
| Optimizer | Adam |
| Batch size | 32 |
| Learning rate (Micro planes $T_i^{(\mathrm{mic})}$) | $10^{-2}$ |
| Learning rate (Renderer $R_\alpha$) | $10^{-2}$ |
| Learning rate (Weights $W_i$) | $10^{-2}$ |
| Learning rate (Base planes $B_k$) | $10^{-2}$ |
| Scheduler | Exponential decay |
| Decay factor | 0.941 |
| Optimization (Training) | |
| Optimizer | Adam |
| Batch size | 32 |
| Learning rate (decoder) | $10^{-4}$ |
| Learning rate (Micro planes $T_i^{(\mathrm{mic})}$) | $10^{-3}$ |
| Learning rate (Renderer $R_\alpha$) | $10^{-3}$ |
| Learning rate (Weights $W_i$) | $10^{-2}$ |
| Learning rate (Base planes $B_k$) | $10^{-2}$ |
| Scheduler | Exponential decay |
| Decay factor | 0.941 |

---

**Algorithm 1** Training a large set of scenes.

---

1: **Input:** $\mathcal{O}, N, N_1, V, E_\phi, D_\psi, \mathcal{R}_\alpha, N_{\text{epoch}}^{(1)}, N_{\text{epoch}}^{(2)}, N_{\text{epoch}}^{(\text{WU})}, \lambda^{(\text{latent})}, \lambda^{(\text{RGB})}, \lambda^{(\text{ae})}, \text{optimizer}$
2: **Random initialization:** $\mathcal{T}^{\text{mic}}, W, \mathcal{B}$
3:
4: *// First $N_1 = 500$ objects (regime 1)*
5: **for** $N_{\text{epoch}}^{(1)}$ steps **do**
6:     **for** $(i,j)$ in shuffle($[\![1, N_1]\!] \times [\![1, V]\!]$) **do**
7:         *// Compute Micro-Macro Planes*
8:         $T_i^{(\text{mic})}, T_i^{(\text{mac})} \leftarrow \mathcal{T}^{(\text{mic})}[i], \; W_i \mathcal{B}$
9:         $T_i \leftarrow T_i^{(\text{mic})} \oplus T_i^{(\text{mac})}$
10:         *// Encode, Render & Decode*
11:         $x_{i,j}, c_{i,j} \leftarrow \mathcal{O}[i][j]$
12:         $z_{i,j} \leftarrow E_\phi(x_{i,j})$
13:         $\tilde{z}_{i,j} \leftarrow \mathcal{R}_\alpha(T_i, c_{i,j})$
14:         $\hat{x}_{i,j}, \tilde{x}_{i,j} \leftarrow D_\psi(z_{i,j}), D_\psi(\tilde{z}_{i,j})$
15:         *// Compute losses*
16:         $L_{i,j}^{(\text{latent})} \leftarrow \|z_{i,j} - \tilde{z}_{i,j}\|_2^2$
17:         $L_{i,j}^{(\text{RGB})} \leftarrow \|x_{i,j} - \tilde{x}_{i,j}\|_2^2$
18:         $L_{i,j}^{(ae)} \leftarrow \|x_{i,j} - \hat{x}_{i,j}\|_2^2$
19:         $L_{i,j} \leftarrow \lambda^{(\text{latent})} L_{i,j}^{(latent)} + \lambda^{(\text{RGB})} L_{i,j}^{(RGB)} + \lambda^{(\text{ae})} L_{i,j}^{(ae)}$
20:         *// Backpropagate*
21:         $T_i^{(\text{mic})}, W_i, \mathcal{B}, \alpha, \phi, \psi \leftarrow \text{optimizer.step}(L_{i,j})$
22:     **end for**
23: **end for**
24:
25: *// Remaining objects (regime 2)*
26: $E_\phi.\text{freeze}()$
27: epoch=1
28: **for** $N_{\text{epoch}}^{(2)}$ steps **do**
29:     **for** $(i,j)$ in shuffle($[\![N_1 + 1, N]\!] \times [\![1, V]\!]$) **do**
30:         *// Compute Micro-Macro Planes*
31:         $T_i^{(\text{mic})}, T_i^{(\text{mac})} \leftarrow \mathcal{T}^{(\text{mic})}[i], \; W_i \mathcal{B}$
32:         $T_i \leftarrow T_i^{(\text{mic})} \oplus T_i^{(\text{mac})}$
33:         *// Encode, Render & Decode*
34:         $x_{i,j}, c_{i,j} \leftarrow \mathcal{O}[i][j]$
35:         $z_{i,j} \leftarrow E_\phi(x_{i,j})$
36:         $\tilde{z}_{i,j} \leftarrow \mathcal{R}_\alpha(T_i, c_{i,j})$
37:         $\tilde{x}_{i,j} \leftarrow D_\psi(\tilde{z}_{i,j})$
38:
39:         **if** epoch $\leq N_{\text{epoch}}^{(\text{WU})}$ **then**
40:             *// Warm-up*
41:             $L_{i,j}^{(\text{latent})} \leftarrow \|z_{i,j} - \tilde{z}_{i,j}\|_2^2$
42:             $T_i^{(\text{mic})}, W_i, \mathcal{B}, \alpha \leftarrow \text{optimizer.step}(L_{i,j}^{(\text{latent})})$
43:         **else**
44:             *// Training*
45:             $L_{i,j}^{(\text{RGB})} \leftarrow \|x_{i,j} - \tilde{x}_{i,j}\|_2^2$
46:             $T_i^{(\text{mic})}, W_i, \mathcal{B}, \alpha, \psi \leftarrow \text{optimizer.step}(L_{i,j}^{(\text{RGB})})$
47:         **end if**
48:     **end for**
49:     epoch $\leftarrow$ epoch $+ 1$
50: **end for**

---

