# OpenReview forum: "Fused-Planes: Why Train a Thousand Tri-Planes When You Can Share?"
_ICLR.cc/2026/Conference — ICLR 2026 Poster_

### Official Review · Reviewer_NFsJ · 2025-10-31

**Soundness:** 3
**Presentation:** 3
**Contribution:** 3
**Rating:** 6
**Confidence:** 3

**Summary:**

This paper proposes Fused-Planes, a tri-planar scene representation for efficiently training large collections of 3D object models. Specifically, each object’s planes are decomposed into an object-specific micro-plane and a class-level macro-plane. The training is conducted in a jointly learned 3D-aware latent space, supervised by both the latent and RGB domains. Experiments on ShapeNet and Basel Faces show that Fused-Planes achieves faster training speed and lower per-object memory than standard Tri-Planes, while maintaining comparable rendering quality.

**Strengths:**

1. This paper tackles a clear challenge in scaling object-centric Tri-Plane models and introduces both micro- and macro-decomposition to exploit inter-object redundancy.

2. The proposed method uses a base-plane bank and a learned weighting mechanism to provide a compact and interpretable way to model class-level structures without sacrificing the planar format useful for 2D backbones.

3. The performance is impressive. The proposed method significantly reduces the training consumption, including time and space overhead, while still maintaining comparable reconstruction performance.

**Weaknesses:**

1. The memory of each object excludes the shared network components, including encoder, decoder, and base planes, which may lead to an unfair comparison with baselines. It would be better if the authors could provide a more balanced total-cost comparison across different values of N.

2. The method is class-specific, which means it cannot be generalized to unknown classes.

3. The evaluation is on object-centric, bounded scenes, e.g., ShapeNet categories and Basel Faces. The proposed method struggles with fine details and unbounded scenes, limiting its applicability relative to non-planar methods.

**Questions:**

See weaknesses

---

> ### Author Response · Authors · 2025-11-20
>
> We thank the reviewer for their review and remarks.  We appreciate that the reviewer found our work to be tackling a **clear challenge**, with **impressive performance**.
>
> We address the reviewer's concerns below.
>
> Please note that we provide additional animated results on the following anonymous link: https://anon-supp.github.io/
>
> ### (w.1) Total cost comparison across different values of N
>
> The reviewer is asking for a comparison of the total costs of our method and its baselines across different values of N.
>
> In our main paper, Figure 4 compares the total memory footprint and total training time for our method and **planar** baselines, across different values of N. We provide below these comparisons for our planar and non-planar baselines.
>
> **Table: Total training time (in days) across different values of N**
>
> |               | Planar | N=1000 | N=2000 | N=5000 | N=10000 | N=20000 |
> |---------------------|:------:|-------:|-------:|-------:|--------:|--------:|
> | **Vanilla-NeRF**        | ✗ | 442.2 | 884.4 | 2211.1 | 4422.2 | 8844.4 |
> | **Instant-NGP**         | ✗ | 5.2   | 10.4  | 26.1   | 52.2   | 104.4  |
> | **TensoRF**             | ✗ | 47.9  | 95.7  | 239.3  | 478.7  | 957.4  |
> | **3DGS**                | ✗ | 6.5   | 13.0  | 32.5   | 65.1   | 130.1  |
> | **K-Planes**            | ✓ | 52.3  | 104.7 | 261.6  | 523.3  | 1046.5 |
> | **Tri-Planes**          | ✓ | 44.7  | 89.3  | 223.3  | 446.7  | 893.3  |
> ||||||||
> | **Fused-Planes-ULW**    | ✓ | 7.2   | 12.2  | 27.1   | 52.0   | 101.7  |
> | **Fused-Planes**        | ✓ | 8.3   | 14.5  | 33.2   | 64.3   | 126.5  |
>
> **Table: Total memory footprint (in GB) across different values of N**
>
> | Method               | Planar | N=1000 | N=2000 | N=5000 | N=10000 | N=20000 |
> |----------------------|:------:|-------:|-------:|-------:|--------:|--------:|
> | **Vanilla-NeRF**     | ✗      | 21.5   | 43.0   | 107.4  | 214.8   | 429.7   |
> | **Instant-NGP**      | ✗      | 184.7  | 369.4  | 923.5  | 1847.0  | 3694.0  |
> | **TensoRF**          | ✗      | 203.4  | 406.9  | 1017.2 | 2034.4  | 4068.7  |
> | **3DGS**             | ✗      | 27.0   | 54.0   | 135.1  | 270.1   | 540.2   |
> | **K-Planes**         | ✓      | 400.6  | 801.1  | 2002.8 | 4005.6  | 8011.1  |
> | **Tri-Planes**       | ✓      | 1.5    | 2.9    | 7.3    | 14.6    | 29.3    |
> ||||||||
> | **Fused-Planes-ULW** | ✓      | 0.4    | 0.4    | 0.4    | 0.4     | 0.4     |
> | **Fused-Planes**     | ✓      | 0.8    | 1.3    | 2.7    | 5.0     | 9.7     |
>
> As presented, Fused-Planes presents competitive training times, and is the fastest planar method. In terms of memory, Fused-Planes and Fused-Planes-ULW are the most lightweight methods. These tables have been included in the revised paper (Section D.6, Tables 13, 14).

---

> > ### Author Response · Authors · 2025-11-20
> >
> > ### (w.2) Multi-class object reconstruction
> >
> > We address the reviewer's concern regarding the class-specificity of our method below.
> >
> > We provide a new set of experiments in which Fused-Planes and Fused-Planes-ULW are trained on datasets containing objects from multiple classes.
> >
> > **In brief,** Fused-Planes is applicable to multi-class data and continues to outperform Tri-Planes in this setting.
> >
> > We introduce three new datasets that combine two, three, and four object classes. In the table below, the first three rows correspond to single-class training, where a separate Fused-Planes model is trained for each individual class. The remaining rows report the results of multi-class training, where a single Fused-Planes model is trained jointly on multiple classes.
> >
> > For conciseness, we report only PSNR here. The full results, including SSIM and LPIPS, and their anlysis, are provided in the revised paper (Section 4.2, Table 4).
> >
> > | Model | # Classes | Speakers | Sofas | Furniture | Cars |
> > |-|-|-|-|-|-|
> > | **Tri-Planes (RGB)** | 1 | 27.02 | 28.48 | 27.42 | 29.69 |
> > | **Fused-Planes-ULW** | 1 | 29.22 | 29.02 | 29.14 | 28.71 |
> > | **Fused-Planes** | 1 | 29.99 | 30.92 | 30.72 | 30.27 |
> > | | | | | | |
> > | **Fused-Planes-ULW** | 2 | 28.63 | 28.93 | — | — |
> > | **Fused-Planes** | 2 | 30.03 | 30.37 | — | — |
> > | | | | | | |
> > | **Fused-Planes-ULW** | 3 | 29.30 | 29.33 | 29.47 | — |
> > | **Fused-Planes** | 3 | 29.84 | 30.08 | 30.31 | — |
> > | | | | | | |
> > | **Fused-Planes-ULW** | 4 | 28.12 | 28.34 | 28.54 | 27.73 |
> > | **Fused-Planes** | 4 | 29.72 | 29.70 | 29.79 | 29.15 |
> >
> > Note: training speed and memory usage remain unchanged w.r.t. the single-category models for these new experiments.
> >
> > We can draw two conclusions from these results.
> > 1. Fused-Planes applied to multi-category data **still outperforms Tri-Planes** in terms of rendering quality, training time and memory footprint.
> > 2. A minor reduction in quality appears as more classes are included, reflecting the increased scene diversity that shared base planes must capture. Importantly, this effect is small, and the resulting quality remains on par with or above that of Tri-Planes.
> >
> > Overall, while initially designed to capture common structures among similar object, we can conclude that Fused-Planes and Fused-Planes-ULW are not limited to single-class data and can extend effectively to multiple object categories. We sincerely thank the reviewer for suggesting these experiments, which revealed that our initial assumption about Fused-Planes being limited to single-classes was unnecessarily restrictive. We have revised the limitations section of our paper accordingly.
> >
> > ### (w.3) Modeling unbounded scenes
> >
> > The reviewer is concerned about the applicability of Fused-Planes compared to other non-planar method, in the context of large-scale scene reconstruction.
> >
> > Indeed, Fused-Planes inherit the same limitations as Tri-Planes on unbounded scenes, as both rely on a planar architecture. Nonetheless, Tri-Planes have become a widely adopted representation for object-centric 3D reconstruction, where this constraint is not a limiting factor in practice. Given that Fused-Planes provide a substantially more efficient alternative while maintaining the same scope, we expect them to be widely applicable within this well-established application domain.
> >
> > Additionally, compared to non-planar methods, Fused-Planes provides stronger practical applicability for the following two reasons:
> > - Non-planar approaches are typically either substantially slower or require considerably more memory, making them costly to deploy in practice at large scale.
> > - Moreover, the planar architecture of Fused-Planes makes it directly integratable in various image-based pipelines.
> >
> > ---
> >
> > In summary, Fused-Planes:
> > - generalizes to multi-class as seen in new experiments,
> > - provides competitive total-cost comparisons across N through newly added analyses,
> > - retains the practical scope of planar object-centric methods while offering major efficiency gains.
> >
> > We sincerely hope that our response resolves the concerns raised. If the reviewer has other technical points to discuss that may currently prevent them from reconsidering the score, we would be more than happy to discuss further.

---

### Official Review · Reviewer_ymg3 · 2025-10-31

**Soundness:** 3
**Presentation:** 3
**Contribution:** 3
**Rating:** 6
**Confidence:** 3

**Summary:**

This paper presents a resource-efficient tri-planar representation (namely, Fused-Planes) for modelling large collections of 3D objects. The core idea is to decompose each object's representation into object-specific 'micro' planes and shared 'macro' planes (constructed from learned base planes), trained in a 3D-aware latent space. The work addresses a real problem, i.e., the computational cost of training thousands of Tri-Planes, and demonstrates impressive efficiency gains while maintaining quality (7.2× faster training, 3.2× lower memory). The combination of micro-macro decomposition with latent space training is well-motivated, and ablations demonstrate both components are necessary. Experiments and ablations are extensively conducted across multiple baselines and datasets.

**Strengths:**

The major strength of this paper is the significant resource savings (7.2× speed, 3.2× memory reduction vs Tri-Planes). Besides, the ultra-lightweight variant achieves 1875× memory reduction. And it maintains the rendering quality.

**Weaknesses:**

The method only works within a single object class. For multiple classes with large visual variations, you need multiple instances of Fused-Planes. This significantly limits practical applicability. For diverse datasets, the overhead of multiple base plane sets could negate efficiency gains.

For open surfaces and unbounded scenes, this method is still limited like other triplane methods.

Table 4, Fused-Planes (Micro)(latent space without macro planes) performs worse than Tri-Planes. This suggests the latent space itself introduces quality degradation, which is only compensated by the macro planes. This is concerning because it means the latent space is not providing a better representation per se. It's just enabling the sharing mechanism.

During inference, each Fused-Planes requires computing the weighted sum at inference (Eq. 2). What's the computational cost of this operation compared to directly loading a Tri-Plane? For applications requiring real-time rendering, is this overhead acceptable? What are the runtime/FPS at inference time across different methods?

**Questions:**

What happens when you train on the entire ShapeNet dataset, not just per-category? How many base plane sets would be needed? What's the break-even point where efficiency gains disappear?

At what scale (N objects) does M=50 need to increase? Is there a theoretical or empirical relationship between M, N, and object class diversity?

---

> ### Author Response · Authors · 2025-11-20
>
> We thank the reviewer for their review and remarks.  We appreciate that the reviewer found our work addressing a **real problem**, and demonstrating **impressive efficiency gains**, with experiments **extensively conducted** across multiple baselines and datasets.
>
> We address the reviewer's concerns below.
>
> Please note that we provide additional animated results on the following anonymous link: https://anon-supp.github.io/
>
> ### (w.1) Multi-class reconstruction
>
> We provide a new set of experiments in which Fused-Planes and Fused-Planes-ULW are trained on datasets containing objects from multiple classes.
>
> **In brief,** Fused-Planes is applicable to multi-class data and continues to outperform Tri-Planes in this setting.
>
> We introduce three new datasets that combine two, three, and four object classes. In the table below, the first three rows correspond to single-class training, where a separate Fused-Planes model is trained for each individual class. The remaining rows report the results of multi-class training, where a single Fused-Planes model is trained jointly on multiple classes.
>
> For conciseness, we report only PSNR here. The full results, including SSIM and LPIPS, and their analysis, are provided in the revised paper (Section 4.2, Table 4).
>
> | Model | # Classes | Speakers | Sofas | Furniture | Cars |
> |-|-|-|-|-|-|
> |**Tri-Planes (RGB)**|1|27.02|28.48|27.42|29.69|
> | **Fused-Planes-ULW** |1|29.22|29.02|29.14|28.71|
> | **Fused-Planes** |1|29.99|30.92|30.72|30.27|
> |||||||
> | **Fused-Planes-ULW** | 2 | 28.63 | 28.93 | — | — |
> | **Fused-Planes** | 2 | 30.03 | 30.37 | — | — |
> |||||||
> | **Fused-Planes-ULW** | 3 | 29.30 | 29.33 | 29.47 | — |
> | **Fused-Planes** | 3 | 29.84 | 30.08 | 30.31 | — |
> |||||||
> | **Fused-Planes-ULW** | 4 | 28.12 | 28.34 | 28.54 | 27.73 |
> | **Fused-Planes** |4|29.72|29.70|29.79|29.15|
>
> Note: training speed and memory usage remain unchanged w.r.t. the single-category models for these new experiments.
>
> We can draw two conclusions from these results.
> 1. Fused-Planes applied to multi-category data **still outperforms Tri-Planes** in terms of rendering quality, training time and memory footprint.
> 2. A minor reduction in quality appears as more classes are included, reflecting the increased scene diversity that shared base planes must capture. Importantly, this effect is small, and the resulting quality remains on par with or above that of Tri-Planes.
>
> Overall, while initially designed to capture common structures among similar object, we can conclude that Fused-Planes and Fused-Planes-ULW are not limited to single-class data and can extend effectively to multiple object categories. We sincerely thank the reviewer for suggesting these experiments, which revealed that our initial assumption about Fused-Planes being limited to single-classes was unnecessarily restrictive. We have revised the limitations section of our paper accordingly.
>
>
> ### (w.2) Tri-Planes' limitation on unbounded scenes
>
> Indeed, Fused-Planes inherit the same limitations as Tri-Planes on unbounded scenes, as both rely on a planar architecture. However, as motivated in the paper, Tri-Planes have become a widely adopted representation for object-centric 3D reconstruction, where this constraint is not a limiting factor in practice. Given that Fused-Planes provide a substantially more efficient alternative while maintaining the same scope, we expect them to be widely applicable within this well-established application domain.

---

> > ### Author Response · Authors · 2025-11-20
> >
> > ### (w.3) Ablation of the Macro planes
> >
> > As the reviewer accurately points out, the ablation in which Fused-Planes is applied in the latent space *without* shared parameters performs worse than Tri-Planes. However, this experiment reflects a configuration that is not representative of our full method. Below, we clarify why the latent space is a deliberate and beneficial design choice in Fused-Planes.
> >
> > #### Why we use a latent space in Fused-Planes
> >
> > In designing Fused-Planes, we have decided to adopt a latent space for the following two reasons:
> > - **Efficiency.** Rendering NeRF-based representations in a latent space leads to substantial savings in training time compared to the RGB space. This is thanks to the reduced resolutions, which alleviate the volume rendering bottleneck.
> > - **Enabling shared components.** Importantly, the latent space enables us to implement shared components in our Fused-Planes representations, which enable our memory savings. In fact, when comparing our ablation "Fused-Planes (RGB)" to Tri-Planes, we can see that using shared components in a standard RGB space leads to quality losses. This is not the case when using a latent space (i.e. when comparing "Fused-Planes (Micro)" with Fused-Planes). Our intuition is that the latent space makes it easier for the shared components to find common structures, and hence provides better representations for Fused-Planes.
> >
> > Additionally, when a latent space is trained jointly with shared representations, it does not lead to quality degradation. In fact, it is quite the contrary for Fused-Planes: the latent space provides better representations that are more effectively learned with shared components.
> >
> > All in all, the latent space does provide a better representation in the context of Fused-Planes, and should not be a concern.
> >
> > ### (w.4) Computational cost of computing Fused-Planes and rendering speed
> >
> > The reviewer raises a concern about the computational overhead of computing Fused-Planes and its potential impact on rendering speed. Below, we provide two comparative tables showing that (i) the overhead is negligible and (ii) Fused-Planes achieve significantly better rendering speeds compared to its planar baselines.
> >
> > #### Overhead analysis (eq. 2)
> >
> > The table below reports the cost of computing a Fused-Planes representation from the base planes and learned coefficients (Eq. 2).
> >
> > |Method|Compute Fused-Planes (eq 2, once)| Render 1 Frame | Render 30s Video @ 30fps |
> > |-|-|-|-|
> > | **Fused-Planes** | 0.65 ms | 10.95 ms | 9.85 s |
> > | **Triplanes (RGB)** |-| 23.30 ms | 20.97 s |
> >
> > As displayed in the table, the overhead indroduced by this computation is largely negligeable compared to rendering.
> >
> > #### Rendering Speed (FPS)
> >
> > The table below reports the rendering FPS of our method and the baselines, using the Nerfstudio reference implementation for all baseline models.
> >
> > | Method | FPS |
> > |-|-|
> > | **Fused-Planes** | 91.3 |
> > | **Triplanes (RGB)** | 42.9 |
> > | **K-Planes** | 14.3 |
> > |||
> > | **INGP** | 48.7 |
> > | **VanillaNeRF** | 0.85 |
> > | **TensoRF** | 13.6 |
> > | **3DGS** | 176.0|
> >
> > As presented, Fused-Planes showcases significantly faster rendering speed than all baselines except 3DGS.
> >
> > Note that in both tables, the reported rendering time for Fused-Planes includes (i) the volume rendering step and (ii) the decoding step that converts the rendered latent representation into an RGB image.
> >
> > These new results have been added to the revised paper (Section D.4, Tables 10 and 11).

---

> > > ### Author Response · Authors · 2025-11-20
> > >
> > > ### (Questions) Relationship between M, N, and class diversity
> > >
> > > The reviewer is asking about relationships between M, N, and class diversity. To answer these questions, we have conducted multiple sets of experiments.
> > >
> > > #### Choice of the value of M
> > >
> > > The first set of experiments studies the impact of the value of M on the performances of Fused-Planes, for which the results are presented below. The table shows that $M=50$ is the optimal value for Fused-Planes.
> > >
> > > ||$M$|Per-Scene Training Time (min)|Total Memory for 2000 scenes (MB)|PSNR|
> > > |-|-|-|-|-|
> > > |Fused-Planes (M=5)|5|8.60|1276|29.89|
> > > |Fused-Planes (M=20)|20|8.61|1291|30.02|
> > > |Fused-Planes|50|8.92|1322|30.27|
> > > |Fused-Planes (M=75)|75|8.99|1348|29.62|
> > >
> > > #### Relationship between M and class diversity
> > >
> > > In order to assess the relationship between M and class diversity, we conduct experiments where we increase the number of classes.
> > >
> > > Specifically, we train Fused-Planes on 1 and 4 classes, with M=50 and M=75.
> > >
> > > |**$M$**|**single-class (cars)**|**single-class (speakers)**|**four classes (cars)**|**four classes (speakers)**|
> > > |-|-|-|-|-|
> > > |**50**|**30.27**|**29.99**|**29.15**|**29.72**|
> > > |**75**|29.62|29.46|28.58|29.20|
> > >
> > > *Note: The table above report PSNR values for NVS evaluation, averaged over multiple scenes of the same category (cars and speakers), when jointly learning one or four classes.*
> > >
> > > Two conclusions can be drawn from these experiments:
> > > - Fused-Planes shows similar performances when increasing the number of classes, indicating that class diversity can be effectively captured with 50 base planes.
> > > - Increasing M beyond 50 does not provide additional benefit.
> > >
> > > This indicates that 50 base planes are sufficient to capture the diversity across a single and multiple classes. These new results have been added to the revised paper (Section D.3, Table 9).
> > >
> > > In summary, these findings provide empirical evidence that (i) M = 50 is an effective and efficient choice for Fused-Planes, and (ii) increasing class diversity does not require scaling M.
> > >
> > > #### Relationship between M and N
> > >
> > > Intuitively, one might expect M to grow with N, since more scenes introduce greater variability. However, as seen in our experiments, M=50 planes effectively captures class diversity even across different classes. As such, increasing N within a single class does not require scaling M.
> > >
> > > #### Questions
> > >
> > > Given these results, we now answer the reviewer's individual questions below.
> > >
> > > > What happens when you train on the entire ShapeNet dataset, not just per-category?
> > >
> > > As shown in our experiments on multi-class training, Fused-Planes showcases similar quality when extending our training from single to multiple classes, with no significant quality degradations when increasing the number of classes. As such, we expect a similar behavior when training on more ShapeNet categories.
> > >
> > > > How many base plane sets would be needed? At what scale (N objects) does M=50 need to increase?
> > >
> > > As shown in our experiments, M=50 base planes is the optimal number to effectively capture common structures in classes, when evaluating on one to four object classes. M does not need to increase when increasing class diversity.
> > >
> > > > What's the break-even point where efficiency gains disappear?
> > >
> > > In terms of memory costs, it is important to keep M << N in order to see memory gains. If M is of the same order of magnitude as N, the memory gains would disappear as the number of trainable features per-object would be similar to that of Tri-Planes. For reference, all of our main experiments have been conducted with M=50 and N=2000.
> > >
> > > In terms of training time, increasing M does not lead to significant slow-downs, as the main gains come from the jointly-trained 3D-aware latent space that alleviate the rendering bottleneck.
> > >
> > >
> > > > Is there a theoretical or empirical relationship between M, N, and object class diversity?
> > >
> > > In summary, our experiments demonstrate that M = 50 remains an effective choice, independent of both class diversity and N. Increasing N only introduces additional class diversity, which, based on our results, does not necessitate increasing M. Thus, M is decoupled from both class diversity and N, on the object classes we have tested. The efficiency advantages of Fused-Planes remain even as class diversity increases.
> > >
> > > ---
> > > In summary, Fused-Planes:
> > > - generalizes to multi-class datasets as seen in new experiments,
> > > - uses a latent space that provides an effective representation,
> > > - presents improved rendering speeds compared to planar baselines,
> > > - requires only M=50 base planes, independently of N and class diversity.
> > >
> > > We sincerely hope that our response resolves the concerns raised. We especially appreciate the reviewer’s constructive suggestion to evaluate on multi-class data, which annulled a limitation of our method and further strengthened our contribution.
> > >
> > > If the reviewer has other technical points to discuss that may currently prevent them from reconsidering the score, we would be more than happy to discuss further.

---

### Official Review · Reviewer_C24d · 2025-11-06

**Soundness:** 3
**Presentation:** 2
**Contribution:** 2
**Rating:** 2
**Confidence:** 4

**Summary:**

This paper introduces Fused-Planes, an efficient tri-plane–based method for reconstructing large classes of 3D objects. The approach incorporates a Micro component to capture object-specific features and a Macro component to encapsulate structural similarities shared across the object class. In addition, it leverages a 3D-aware latent space to accelerate both the rendering and training processes of Fused-Planes. Compared with the original Tri-Planes, Fused-Planes achieves 7.2× faster training and 3.2× lower memory consumption while maintaining comparable rendering quality. The authors further propose an ultra-lightweight variant that almost entirely omits the micro component, yielding a remarkable 1875× memory reduction. Experimental results demonstrate that Fused-Planes significantly outperforms existing plane-based methods such as Tri-Planes and K-Planes in terms of both training efficiency and memory scalability for large-scale multi-object reconstruction.

**Strengths:**

1. The paper introduces a 3D aware latent space as a form of shared representation in the object reconstruction domain, enabling the model to better capture structural similarities across object classes. According to the ablation study, employing this latent representation rather than directly optimizing in RGB space leads to faster convergence while maintaining comparable rendering quality.
2. Compared to C3 NeRF, which scales only up to around 20 scenes, the proposed approach remains scalable to thousands of objects, demonstrating superior generalization and efficiency across large datasets.
3. Under the current task setting, the method achieves significant advantages in training speed and memory efficiency over traditional KPlanes and TriPlanes approaches.

**Weaknesses:**

1. Although the paper claims that Fused-Planes remains scalable to thousands of objects, I do not observe convincing evidence of this property from the presented experimental results or supplementary videos.
2. In line 103, the paper states that TensoRF, 3DGS, and Instant-NGP cannot be reshaped into image-like tensors. However, to my knowledge, several recent works in the 3DGS domain, such as Animatable Gaussians, ASH, GaussianAvatar, and Reperformer, have successfully employed 2D UV unwrapping or Morton mapping strategies to project 3D point clouds into 2D grids and then apply image-based CNN architectures to learn the appearance of avatars under novel motions. I believe these 2D parameterization methods should be properly cited and discussed for completeness.
3. While I understand the limitations of Tri-Planes in representing fine details and handling unbounded scenes, the object-centric datasets used in this paper (e.g., ShapeNet, Basel Faces) appear to be extremely simple in geometry. Despite such simplicity, both Fused-Planes and Fused-Planes-ULW still produce noticeably blurry renderings. This level of visual quality makes it difficult to assess the practical value and applicability of the proposed approach.
4. To my knowledge, several tri-plane-based methods such as TeTriRF can generate highly detailed and realistic renderings of complex human data with relatively lightweight models. It is therefore unclear why the proposed method fails to achieve comparable quality even on synthetic datasets with simple geometries.
5. Furthermore, in Fig. 5, where Fused-Planes is compared with other per-scene training methods in terms of size and training time, I believe this comparison is highly unfair

**Questions:**

As noted in the weaknesses section, the paper should better highlight its capability to scale up to thousands of objects. It should also provide a discussion of relevant references regarding the use of 2D parameterization in 3DGS, and clarify the factors contributing to its lower rendering quality.

**Details Of Ethics Concerns:**

No ethics concerns of this paper.

---

> ### Author Response · Authors · 2025-11-20
>
> We thank the reviewer for their helpful comments and remarks. We appreciate that the reviewer found the performances achieved by our method to be **remarkable**, with **significant advantages** in training speed and memory efficiency over traditional approaches.
>
> We would like to clarify two crucial points before addressing individual concerns in more detail.
> - Fused-Planes is indeed a large-scale method. In the paper, we only show subsets of objects from our adopted datasets due to space constraints. In practice, we have reconstructed 2000 objects for each dataset, for a total of 10000 reconstructed objects across our main expriments. The following anonymous link shows a larger subset of 1000 objects: https://anon-supp.github.io/ .
> - Fused-Planes achieves rendering quality comparable to or better than Tri-Planes. Hence, the quality achieved by Fused-Planes is on-par with other planar methods. This should resolve any concerns related to quality. Comparisons to other single-scene methods are included only to contextualize our approach, not as the primary benchmark.
>
> In light of the clarifications provided above, and detailed below, we respectfully invite the reviewer to reconsider their assessment.
>
> We address the reviewer's individual concerns below.
>
> ### (w.1) Evidence of scaling to thousands of objects
>
> We respectfully clarify a misunderstanding in the review. The reviewer states that our method does not demonstrate evidence of scaling to thousands of objects; however, as described in Section 4, all our experiments were performed on N = 2000 objects.
>
> Due to space constraints, the main paper and appendix illustrate only a representative subset of these results. Specifically, in Figures 7 to 11, we show 4 objects per dataset and per representation, amounting to 200 illustrated objects.
>
> As illustrated in Figure 4, our method presents favorable resource efficiency as the total number of objects increases, and significant improvements compared to our baselines for N=2000 objects (our experimental setting).
>
> For a more comprehensive visualization, we provide animated renderings of 1000 objects reconstructed with Fused-Planes in the supplementary material and in the following anonymous link: https://anon-supp.github.io . We have also added this visualization in the revised paper (Figure 10).
>
> We hope this resolves the reviewer’s concern regarding the evidence of scalability of our method.
>
> ### (w.2) Works mapping 3DGS to planes
>
> We thank the reviewer for bringing these works to our attention. Following the reviewer’s suggestion, we have modified the related work of our paper to discuss them (see paper revision, in blue).
>
> In summary, these works do make it possible to combine 3D Gaussian Splatting with CNN-based architectures. They do so by introducing 2D parameterizations of 3DGS, typically through additional unwrapping or mapping procedures that convert 3D Gaussians into 2D Gaussian maps. Within this line of research, 3DGS has become a natural choice for tasks involving scene deformation or animation.
>
> Tri-Planes, by contrast, offer a fixed and structured 2D tensor representation that aligns directly with standard image-based architectures, requiring no extra mapping steps. This structural compatibility is a major reason for their broad adoption in large-scale 3D reconstruction pipelines [1,2], and it motivates our focus on planar representations.

---

> > ### Author Response · Authors · 2025-11-20
> >
> > ### (w.3 + w.4) Adopted datasets and rendering quality of Fused-Planes
> >
> >
> > Fused-Planes requires lower resource costs while showcasing similar or better quality than Tri-Planes, while preserving their full scope of use, hence demonstrating practical value.
> >
> > #### Adopted datasets
> >
> > Our dataset choices demonstrates the practical value of our approach in two key ways:
> >
> > - **The chosen datasets are standard for Tri-Planes.** Tri-Planes are mainly used as object-centric 3D representations [1,2,3,4]. As Fused-Planes are intended to improve upon the resource costs of Tri-Planes **while keeping their key planar property**, we evaluate them in the same context as Tri-Planes.
> > - **We evaluate Fused-Planes on the same large-scale 3D datasets used by our multi-scene baselines.** We utilize ShapeNet because it is also used by our primary multi-scene baseline (CodeNeRF). In addition, we evaluate on the Basel Face dataset to offer a broader assessment of the method across different object-centric domains.
> >
> >
> > #### Rendering quality
> >
> > The reviewer states that Fused-Planes and Fused-Planes-ULW produce blurry renderings, hence questioning practical applicability. We would like to clarify:
> > - **In the case of Fused-Planes**, our illustrations (notably in Figure 3) show that Fused-Planes is **sharper** than our Tri-Planes baseline. This is also validated by our evaluation metrics, where Fused-Planes outperform Tri-Planes in rendering quality.
> >     - Compared to Tri-Planes on ShapeNet, Fused-Planes demonstrates a PSNR $(\uparrow)$ of 30.47 vs. 28.15; a SSIM $(\uparrow)$ of 0.957 vs. 0.919, and a LPIPS $(\downarrow)$ of 0.042 vs. 0.121)
> > - **In the case of Fused-Planes-ULW**, we believe that the quality trade-offs are acceptable, as it is an ultra-lightweight variant of Fused-Planes that, as illustrated in our quantitative evaluations, trades-off minor rendering quality (PSNR of 29.02dB instead of 30.47dB) for substantial resource gains ($600 \times$ lighter than Fused-Planes, and $1875 \times$ lighter than Tri-Planes). This model is only proposed as an alternative to Fused-Planes, where resource costs are a top priority.
> >
> > As a result, the rendering quality of Fused-Planes does not undermine its practical applicability.
> >
> > #### Alternative methods utilizing planar architectures
> >
> > The reviewer mentions alternative methods that utilize planar structures, such as TeTriRF, that can reconstruct more detailed 3D scenes.
> >
> > While this is correct, these methods can represent higher details by **abandoning the strict planarity of Tri-Planes** and incorporating auxiliary architectures that explicitly learn 3D geometry. Specifically, TeTriRF uses a voxel grid to model scene geometry, while the Tri-Planes serve primarily as a color feature representation. In this setup, the geometry is learned by the voxel grid and subsequently "painted" using features learned by the Tri-Planes.
> >
> > In designing Fused-Planes, our goal was precisely to avoid introducing auxiliary volumetric components that break the pure planarity of Tri-Planes, which is one of the main reasons they have been such an appealing representation. By preserving pure planarity, Fused-Planes maintains this compatibility while significantly reducing resource costs.
> >
> > #### Concluding on this concern
> >
> > Overall, Fused-Planes provides similar or better quality than its Tri-Planes baselines, while remaining purely planar, and significantly reducing upon resource costs in large scale settings. We hope that our message clarifies the reviewer's concern.
> >
> > ### (w.5) Comparison to other per-scene methods
> >
> > The reviewer notes that the comparison with per-scene training methods in Figure 5 may be unfair. We would like to clarify that this comparison is intended to illustrate the practical landscape of available strategies for reconstructing a large number of 3D objects.
> >
> > We adopt the perspective of a practitioner who wishes to train a thousand object representations today and outline the options they would consider: (1) training many per-scene methods, (2) using a multi-scene approach such as CodeNeRF, or (3) employing Fused-Planes. In such a case, Fused-Planes is the most favorable method in terms of resource efficiency.
> >
> > To avoid any misinterpretation, the revised version of the paper now explicitly states this intention in the caption of Figure 5. We thank the reviewer for pointing out this possible source of confusion.
> >
> >
> > ---
> > In summary, Fused-Planes:
> > - reconstructs thousands of objects efficiently,
> > - is evaluated on datasets that are standard for Tri-Planes, and on which it exceeds Tri-Planes in rendering quality,
> > - offers state-of-the-art training speed and memory efficiency among planar methods,
> >
> > We sincerely hope that our response resolves the concerns raised. If the reviewer has other technical points to discuss that may currently prevent them from reconsidering the score, we would be more than happy to discuss further.

---

> > > ### Author Response · Authors · 2025-11-20
> > >
> > > ### References
> > >
> > > [1] Shue, J. R., Chan, E. R., Po, R., Ankner, Z., Wu, J., & Wetzstein, G. (2023, June). 3D Neural Field Generation Using Triplane Diffusion. Proceedings of the IEEE/CVF Conference on Computer Vision and Pattern Recognition (CVPR), 20875–20886.
> > >
> > > [2] Liu, Y.-T., Guo, Y.-C., Luo, G., Sun, H., Yin, W., & Zhang, S.-H. (2024, June). PI3D: Efficient Text-to-3D Generation with Pseudo-Image Diffusion. Proceedings of the IEEE/CVF Conference on Computer Vision and Pattern Recognition (CVPR), 19915–19924.
> > >
> > > [3] Wang, T., Zhang, B., Zhang, T., Gu, S., Bao, J., Baltrusaitis, T., … Guo, B. (2023, June). RODIN: A Generative Model for Sculpting 3D Digital Avatars Using Diffusion. Proceedings of the IEEE/CVF Conference on Computer Vision and Pattern Recognition (CVPR), 4563–4573.
> > >
> > > [4] Anciukevičius, T., Xu, Z., Fisher, M., Henderson, P., Bilen, H., Mitra, N. J., & Guerrero, P. (2023, June). RenderDiffusion: Image Diffusion for 3D Reconstruction, Inpainting and Generation. Proceedings of the IEEE/CVF Conference on Computer Vision and Pattern Recognition (CVPR), 12608–12618.

---

> > > > ### Comment · Reviewer_C24d · 2025-11-21
> > > > **Response to rebuttal**
> > > >
> > > > Thank you very much for the authors’ detailed reply. The supplementary results provided at https://anon-supp.github.io
> > > > , together with the clarification in w.5, have effectively resolved several of my concerns. The revision of the Fig. 5 caption, as well as the additional discussion on Gaussian-plane parameterization methods, further highlights the advantages of your approach. I also believe that the paper could explicitly emphasize that Fused-Planes preserves the purely planar representation and compatibility of Tri-Planes without introducing extra components. Consequently, its rendering quality may be slightly lower than methods such as TeTriRF that rely on richer geometric structures. Nevertheless, the rebuttal has addressed most of my questions, and I am therefore raising my score to 6.

---

> > > > > ### Author Response · Authors · 2025-11-25
> > > > >
> > > > > We sincerely thank the reviewer for their feedback and for reassessing our contributions following the revised submission. In response, we have made an additional minor revision, highlighted in magenta in the updated manuscript, which further clarifies our focus on pure planarity and our avoidance of additional components such as those used in TetriRF. We remain open for any further discussion.

---

### Official Review · Reviewer_QUGX · 2025-11-08

**Soundness:** 3
**Presentation:** 3
**Contribution:** 3
**Rating:** 6
**Confidence:** 3

**Summary:**

This paper proposes Fused-Planes, a shared tri-planar representation that improves the efficiency of training large 3D object collections. Instead of training independent Tri-Planes for each object, the method decomposes each object’s planes into a micro component (object-specific details) and a macro component (a weighted sum of shared base planes capturing class-level structure). The model is trained jointly within a 3D-aware latent space, which further reduces computation. Experiments on ShapeNet and Basel Faces show strong results: up to 7.2× faster training, 3.2× smaller per-object memory, and comparable or better rendering quality than Tri-Planes. An ultra-lightweight variant achieves extreme compression with minor quality loss.

**Strengths:**

* **Clear and practical contribution.** The paper tackles a real inefficiency in Tri-Plane training and offers an intuitive solution that effectively shares structure within a class.
* **Strong empirical gains.** Training speed, memory footprint, and quality all improve substantially. The ULW version demonstrates impressive compression with minimal degradation.
* **Elegant design.** The micro–macro split is simple yet powerful, allowing reuse of planar architectures while amortizing cost across objects.
* **Comprehensive evaluation.** The experiments, ablations, and comparisons are thorough and well-presented. The results convincingly support the claims.
* **Readable and well-organized.** The paper is clear, figures are informative, and the setup is easy to follow.

**Weaknesses:**

* **Single-class limitation.** The method assumes one class per model. Scaling to diverse datasets would require separate sets of base planes, reducing its flexibility.
* **Limited analysis of shared bases.** The learned base planes are not explored in depth. It is unclear what structures they capture or how weights vary across instances.
* **Restricted evaluation scope.** Experiments focus only on novel view synthesis of synthetic datasets. No tests on real or multi-class data, or on downstream tasks that might use the planar outputs.
* **Dependence on latent space.** The model relies on a pretrained VAE initialization, but the sensitivity to this setup and its cost are not fully studied.
* **Constant overhead.** While per-object cost drops sharply, the shared components introduce a large fixed memory load, which only becomes efficient at larger scales.

**Questions:**

1. How sensitive is performance to the number of shared base planes and the micro–macro feature split?
2. Can a single set of base planes handle multiple classes, perhaps with class-conditioned weights?
3. What do the shared base planes actually learn? Visualizing or analyzing them could offer useful insight.
4. How dependent is training on the pretrained latent codec? Could the system work with a smaller or scratch-trained encoder?
5. Could Fused-Planes be tested on real multi-view datasets or downstream 2D-compatible tasks to demonstrate broader utility?

---

> ### Author Response · Authors · 2025-11-20
>
> We thank the reviewer for their review and remarks. We appreciate that the reviewer found our contribution to be **clear and practical**, with **strong emprical gains**, an **elegant design**, and containing a **comprehensive evaluation**.
>
> We address the reviewer's concerns below.
>
> ### (w.1 + q.2) Evaluation on multiple classes
>
> We provide a new set of experiments in which Fused-Planes and Fused-Planes-ULW are trained on datasets containing objects from multiple classes.
>
> **In brief,** Fused-Planes is applicable to multi-class data, without requiring multiple sets of base planes, and continues to outperform Tri-Planes in this setting.
>
> We introduce three new datasets that combine two, three, and four object classes. In the table below, the first three rows correspond to single-class training, where a separate Fused-Planes model is trained for each individual class. The remaining rows report the results of multi-class training, where a single Fused-Planes model is trained jointly on multiple classes.
>
> For conciseness, we report only **PSNR** here. The full results, including SSIM and LPIPS, and its analysis, are provided in the revised paper (Section 4.2, Table 4).
>
> | Model | # Classes | Speakers | Sofas | Furniture | Cars |
> |-|-|-|-|-|-|
> | **Tri-Planes (RGB)** | 1 | 27.02 | 28.48 | 27.42 | 29.69 |
> | **Fused-Planes-ULW** | 1 | 29.22 | 29.02 | 29.14 | 28.71 |
> | **Fused-Planes** | 1 | 29.99 | 30.92 | 30.72 | 30.27 |
> | | | | | | |
> | **Fused-Planes-ULW** | 2 | 28.63 | 28.93 | — | — |
> | **Fused-Planes** | 2 | 30.03 | 30.37 | — | — |
> | | | | | | |
> | **Fused-Planes-ULW** | 3 | 29.30 | 29.33 | 29.47 | — |
> | **Fused-Planes** | 3 | 29.84 | 30.08 | 30.31 | — |
> | | | | | | |
> | **Fused-Planes-ULW** | 4 | 28.12 | 28.34 | 28.54 | 27.73 |
> | **Fused-Planes** | 4 | 29.72 | 29.70 | 29.79 | 29.15 |
>
> Note: training speed and memory usage remain unchanged w.r.t. the single-category models for these new experiments.
>
> We can draw two conclusions from these results.
> 1. Fused-Planes applied to multi-category data **still outperforms Tri-Planes** in terms of rendering quality, training time and memory footprint.
> 2. A minor reduction in quality appears as more classes are included, reflecting the increased scene diversity that shared base planes must capture. Importantly, this effect is small, and the resulting quality remains on par with or above that of Tri-Planes.
>
> Overall, while initially designed to capture common structures among similar object, we can conclude that Fused-Planes and Fused-Planes-ULW are not limited to single-class data and can extend effectively to multiple object categories. We sincerely thank the reviewer for suggesting these experiments, which revealed that our initial assumption about Fused-Planes being limited to single-class data was unnecessarily restrictive. We have revised the limitations section of our paper accordingly.
>
> ### (w.2 + q.3) Analysis of shared base planes
>
> We have added a set visualizations and analyses of the shared base planes in the revised paper (Section E) and at the following anonymous link: https://anon-supp.github.io/.
>
> The first set of visualizations illustrates the renderings obtained from individual base planes. The protocol used to obtain these visualizations is explained on the anonymous website and in the revised paper.
>
> **Observation:** These visualizations show that base planes can be grouped into two categories:
> - Semantic: some base planes clearly encode object-level structures (e.g., faces, cars),
> - Residual: other base planes capture finer intra-class variability relative to the semantic base planes.
>
> Together, these base planes contribute to each object representation.
>
> We further visualize the values of the learned weights $W_i$ used to compose each Fused-Planes representation using $T_i = \sum_{k=1}^{M} w_i^{k} B_k$.
>
> **Observation:** Across scenes, a small subset of planes typically dominates the linear combination, while others provide small corrective contributions.
>
> Finally, we also visualize the Fused-Planes resulting from a weight interpolation. The detailed protocol for this visualization is also presented on the anonymous site.
>
> **Observations:** Interpolations between weights yield coherent scenes, where we transition smoothly from one scene to another (e.g. the mouth closes gradually across the different faces).

---

> > ### Author Response · Authors · 2025-11-20
> >
> > ### (w.3 + q.5) Evaluation scope
> >
> > Regarding evaluation on multi-class data, see (w.1 + q.2).
> >
> > Regarding real-world scenes, as discussed in section 4.3, our method is applicable to the same types of scenes as Tri-Planes, as it shares their planar architecture. This includes object-centric scenes, but does not include real-world scenes characterized by unbounded horizons and fine details, which are not compatible with Tri-Planes.
> > We evaluate Fused-Planes on object-centric scenes, on which it demonstrates similar or superior rendering quality compared to Tri-Planes, while significantly improving upon their resource costs.
> >
> > The reviewer also suggests additional evaluations on downstream tasks associated with planar representation. Our contribution focuses on the efficient construction of large sets of Tri-Planes and can be evaluated independently of downstream tasks utilizing the planarity of Fused-Planes. This is because Fused-Planes remains **fully compatible** with architectures that rely on Tri-Planes for downstream tasks, as it preserves their purely planar structure. While we agree that using our representation within other downstream pipelines could further demonstrate Fused-Planes practical utility, such integration lies beyond the scope of this work and is left for future exploration.
> >
> > ### (w.4 + q.4) Dependence on latent space
> >
> > The reviewer is asking about the dependence and sensitivity of Fused-Planes on the pre-training of the adopted VAE.
> >
> > To answer this question, we conduct new experiments where we train Fused-Planes with a VAE that has been reset (all weights are randomly initialized) and trained on our scenes with a low budget. This experiment is conducted on 2000 scenes from ShapeNet Cars.
> >
> > In order to avoid backpropagating random gradients to the modules in Fused-Planes at initialization, we allocate $15 \%$ of training time to warm up the VAE after its reset, using the images of the scenes.  This is necessary as training Fused-Planes with a non-functional VAE makes it impossible for Fused-Planes to learn the scenes.
> >
> > The results indicate that employing a VAE trained on a smaller dataset and with lower budget introduces only minor degradation in output quality. This suggests that our framework exhibits **low sensitivity** to the specific initialization of the VAE.
> >
> > | | Mean PSNR | Mean SSIM | Mean LPIPS |
> > |-|-|-|-|
> > | **Fused-Planes (low-budget VAE)** | 29.22 | 0.953 | 0.035 |
> > | **Fused-Planes** |  30.27 | 0.960 | 0.033 |
> >
> > These new results have been added to the revised paper (Section D.5, table 12)
> >
> > ### (w.5) Constant overhead
> >
> > As the reviewer correctly notes, our method introduces a constant overhead for the shared components, which is illustrated in Figure 4.
> >
> > However, this overhead is acceptable for two main reasons. First, the overhead is fixed but not large: it is smaller than the size of a single K-Planes representation (section D.7), making it negligible in comparison to the per-scene savings. Second, as Fused-Planes is designed for large-scale scenarios, the relative impact of this fixed cost diminishes rapidly as the number of objects increases.
> >
> > For instance, representing 2000 objects
> > - using Fused-Planes requires:
> >     - 360.75 (overhead) + 0.48 x 2000 = $1~322$ MB;
> > - where as doing so with K-Planes requires:
> >     - 0 (overhead) + 410.17 x 2000 = $820~340$ MB ;
> > - and using Tri-Planes, it requires:
> >     - 0 (overhead) + 1.5 x 2000 = $3~200$ MB.
> >
> > Therefore, this overhead is acceptable given the substantial per-object savings achieved at scale.

---

> > > ### Author Response · Authors · 2025-11-20
> > >
> > > ### (q.1) Performance relative to number of shared planes and micro-macro split
> > >
> > > #### Performance relative to the number of shared base planes
> > >
> > > We present below a study of the effect of different values of $M$, the number of shared base planes. We also include these results in the revised version of the paper (Section D.3, Table 9).
> > >
> > > As shown, rendering quality varies only minimally across different values of $M$. We select $M = 50$ because it offers the best quality while maintaining similar training time and memory usage.
> > >
> > > ||M|Per-Scene Training Time (min)|Total Memory for 2000 scenes (MB)|PSNR|
> > > |-|-|-|-|-|
> > > |Fused-Planes ($M=5$)|5|8.60|1276|29.89|
> > > |Fused-Planes ($M=20$)|20|8.61|1291|30.02|
> > > |Fused-Planes|50|8.92|1322|30.27|
> > > |Fused-Planes ($M=75$)|75|8.99|1348|29.62|
> > >
> > > #### Performance of the micro-macro split
> > >
> > > We clarify below the effect of the micro-macro split on the performances of Fused-Planes.
> > >
> > > Regarding rendering quality, the micro–macro split achieves the best PSNR, SSIM, and LPIPS among the evaluated configurations. Importantly, varying this split does not lead to substantial performance degradation; the choice primarily reflects a trade-off between efficiency and resource usage rather than a strong influence on reconstruction fidelity.
> > >
> > > Regarding resource costs, the most lightweight configuration is obtained by omitting the micro planes, which yields the highest efficiency.
> > >
> > > |$F^\mathrm{mic}$|$F^\mathrm{mac}$|PSNR ($\uparrow$)|SSIM ($\uparrow$)|LPIPS ($\downarrow$) |Per-scene Training Time (min)|Total Memory for 2000 scenes (MB)|
> > > |-|-|-|-|-|-|-|
> > > |32|0|27.64|0.941|0.040|12.84|3000|
> > > |10|22|28.64|0.950|0.037|8.92|1320.75|
> > > |0|32|27.51|0.935|0.063|7.16|385.8|
> > >
> > > ---
> > >
> > > In summary,
> > > - Fused-Planes can model multiple classes with a single set of base planes,
> > > - the base planes, which we have further analyzed, showcase interesting semantics,
> > > - our method is evaluated on the same types of scenes as Tri-Planes, where it demonstrates superior performance.
> > > - Fused-Planes shows low sensitivity to the VAE initialization,
> > > - Fused-Planes requires an acceptable overhead in large-scale settings.
> > >
> > > We sincerely hope that our response resolves the concerns raised. We especially appreciate the reviewer’s constructive suggestion to evaluate on multi-class data, which annulled a limitation of our method and further strengthened our contribution.
> > >
> > > If the reviewer has other technical points to discuss that may currently prevent them from reconsidering the score, we would be more than happy to discuss further.

---

### Author Response · Authors · 2025-11-20
**Paper revision**

We thank the reviewers for their constructive remarks. We have addressed each reviewer's concerns individually, and summarize below the changes we have integrated into our paper revision.

Our animated results can be viewed at the following anonymous link: https://anon-supp.github.io/.
In the final paper, these animations will be hosted on a de-anonymized project page.
Non-animated versions of the results are also included in the revised paper.


- **Large-scale results.** We have added a visualization of our large-scale results in Figure 10 (C24d).
- ***New experiments:* Multi-class training.** We have added new experiments on multi-class training (Section 4.2, Table 4) (QUGX, ymg3, NFsJ).
- **Updated limitations.** We have updated the limitations section of our paper to take into account the new results on multi-class training.
- ***New experiments:* Impact of $M$.** We have added an ablation study of the effect of the number of base planes $M$ on Fused-Planes in single and multi-class reconstruction (Section D.3, Table 9) (QUGX, ymg3).
- ***New experiments:* Rendering speed.** We have added an analysis of rendering speed in Section D.4 (Section D.4, Tables 10 and 11). (ymg3).
- ***New experiments:* Training with a low-budget VAE.** We have added experiments done on Fused-Planes using a low-budget AE (Section D.5, table 12) (QUGX).
- ***New experiments:* Analysis of the base planes.** We have added in section E an analysis of the representations learned by the base planes (Figure 7), an analysis of the weights multiplying these base planes (Figure 8). We also added a new experiment in which we interpolate between learn weights (Figure 9) (QUGX).
- **Analysis of the cost of Fused-Planes when $N$ varies.** We have added an analysis of the total cost across different values of $N$ (Section D.6, Tables 13, 14) (NFsJ).
- **Related work.** We have revised the related work to include the works referenced by reviewer C24d.
- **Clarification** We have revised the caption of Figure 5 (C24d).

We hope that our paper revision, along with the individual responses, address the reviewers' concerns. We remain open for further discussion.

---

### Author Response · Authors · 2025-11-27
**Friendly reminder**

We thank the reviewers again for their feedback. As the discussion period is ending soon, we would like to gently remind the reviewers to provide us feedback on the revised version of our paper. We have taken great care in revising it and have addressed all of the reviewers’ concerns.

We remain open for further discussion.

---

### Author Response · Authors · 2025-12-01
**Summary for new Area Chair**

We would like to express our gratitude to the area chair for taking on this responsibility under unexpected and difficult circumstances. We thank them in advance for their time and efforts.

To aid in their assessment, we include a concise summary of our reviews and discussions below.

## Evaluation history

**Initial scores:** 6, 2, 6, 6

**Updated scores:** 6, 6, 6, 6

**Date of change:** 21st of November

**Reason for change:** As explicitly stated in their reply, Reviewer C24d **changed their score from 2 to 6**, after we clarified two crucial points which has initially caused a misunderstanding. The reviewer recognizes that our rebuttal has resolved their concerns.

We responded to the questions and remarks of the other reviewers, but we unfortunately did not get any response before the incident occurred.

## Recognized strengths
The reviewers have recognized the following strengths of our contribution:

- Significant gains in training speed and memory footprint *(QUGX, C24d, ymg3, NFsJ)*,
- Does not sacrifice planarity, useful for 2D backbones *(NFsJ)*,
- Scalable to thousands of object *(C24d)*,
- Comprehensive evaluation *(QUGX, ymg3)*,
- Clear and practical contribution *(QUGX, NFsJ)*,
- Clearly written and well-organized *(QUGX, NFsJ)*,
- Elegant design *(QUGX)*.


## Reviewer concerns and our response

We expose below a summary of the reviewers' concerns and our responses. Our new results and modifications are available in our paper revision. Animated results can be found at the following anonymous link (https://anon-supp.github.io/).

### Resolved concerns (confirmed by reviewer)

- **Large-scale results.**
    - Reviewer C24d asked for concrete proof of the large-scale property of our work, as we only visualize a few objects per category in our initial paper.
    - We have provided a visualization of 1000 objects in our revised paper and the anonymous link provided above.
- **Related work and clarifications** Following the reviewer' suggestion, we revised our Related Work section to clarify how our Fused-Planes representation compares to 3DGS–based methods, specifically regarding the compatibility of their architectures with image-based models.


### Addressed concerns (pending confirmation)

- **Multi-class training.**
    - Reviewers *QUGX, ymg3, and NFsJ* had concerns about our method only being tested on single-class datasets.
    - Our new experiments on several multi-class datasets show that Fused-Planes continues to outperform its baselines, with the same efficiency gains.
- **Impact of our hyperparameters.**
    - Reviewers *QUGX and ymg3* asked for experiments on the impact of the number of shared planes $M$.
    - Our new experiments show that rendering quality varies minimally across values of $M$. Our choice of $M$ offers the best quality/memory trade-off.
- **Rendering speed.**
    - Reviewer *ymg3* was concerned about the rendering speed of our method.
    - Our new experiments show that Fused-Planes shows significantly faster rendering speeds compared to Tri-Planes and all other baselines, except 3DGS.
- **Analysis of base planes.**
    - Reviewer *QUGX* asked for a deeper analysis of our base planes.
    - New visualization of these structures are available in our revised paper as well as the provided anonymous link.
    - Our base planes learn interesting common structures.
- **Analysis of the cost of Fused-Planes when the number of objects varies.**
    - In response to reviewer NFsJ, we have added an analysis of the total cost of Fused-Planes across different numbers of objects.
    - Our method showcases competitive training times, and is the fastest planar method.
    - In terms of memory, our method is the most lightweight method.


## Concluding

We sincerely regret that circumstances prevented discussion with the reviewers. We believe we have positively addressed reviewer concerns in our rebuttal, with proofs backed by thorough experiments. We thank the Area Chair for their consideration of these elements.

Sincerely, the authors.

---

### Meta-Review · Area_Chair_VNQz · 2026-01-06

**Summary:**

The rebuttal states that the paper initially received scores of 6, 2, 6, 6, which were updated to 6, 6, 6, 6 after Reviewer C24d confirmed on November 21. This rebuttal effectively addresses key concerns from the reviewers: Reviewer C24d’s main doubt about large-scale applicability is resolved with solid evidence (supplementary visualizations of 1,000 objects), while other issues (multi-class training, hyperparameter sensitivity, rendering speed) are tackled through rigorous new experiments. The paper's strengths (efficiency, planarity, scalability) are reinforced, and supplementary materials enhance transparency. Despite some unconfirmed responses due to external factors, the authors’ thorough revisions and resolution of core issues strongly support the paper’s validity and contribution.

After reviewing all feedback and rebuttals, the AC believes this paper meets the acceptance criteria and recommend it for a poster presentation. The authors are encouraged to integrate all feedback into the final version.

**Reviewer Concerns:**

- Reviewer QUGX: Concerns are well addressed.
- Reviewer C24d: Concerns are well addressed and confirmed by C24d on November 21, who promised to raise the score to 6.
- Reviewer ymg3: Concerns are well addressed.
- Reviewer NFsJ: Concerns are well addressed.

**Reviewer Scores:**

- Reviewer QUGX (conf 3): Likely to maintain the score (6).
- Reviewer C24d (conf 4): Likely to raise the score (6).
- Reviewer ymg3 (conf 3): Likely to maintain the score (6).
- Reviewer NFsJ (conf 3): Likely to maintain the score (6).

---

### Decision · Program_Chairs · 2026-01-26

Accept (Poster)